# Ubiquitin variants potently inhibit SARS-CoV-2 PLpro and viral replication via a novel site distal to the protease active site

**Vera J. E. van Vliet**[1,2☯], **Nhan Huynh**[3,4☯], **Judith Palà**[3,4], **Ankoor Patel**[5], **Alex Singer**[3,4], **Cole Slater**[5], **Jacky Chung**[3,4], **Mariska van Huizen**[1], **Joan Teyra**[3,4], **Shane Miersch**[3,4], **Gia-Khanh Luu**[3,4], **Wei Ye**[3,4], **Nitin Sharma**[6], **Safder S. Ganaie**[6], **Raquel Russell**[5], **Chao Chen**[3,4], **Mindy Maynard**[3,4], **Gaya K. Amarasinghe**[6], **Brian L. Mark**[5]*, **Marjolein Kikkert**[1]*, **Sachdev S. Sidhu**[3,4]*

**1** Department of Medical Microbiology, Leiden University Center of Infectious Diseases (LU-CID), Leiden University Medical Center, Leiden, South Holland, The Netherlands, **2** The Roslin Institute, University of Edinburgh, Midlothian, Scotland, United Kingdom, **3** The Anvil Institute, Kitchener, Ontario, Canada, **4** School of Pharmacy, University of Waterloo, Waterloo, Ontario, Canada, **5** Department of Microbiology, University of Manitoba, Winnipeg, Manitoba, Canada, **6** Department of Pathology and Immunology, Washington University School of Medicine, St. Louis, Missouri, United States of America

☯ These authors contributed equally to this work.
* brian.mark@umanitoba.ca (BLM); m.kikkert@lumc.nl (MK); sachdev.sidhu@gmail.com (SSS)

**Data Availability Statement:** All relevant data are within the manuscript and its Supporting Information files.

## Abstract

The severe acute respiratory syndrome coronavirus 2 (SARS-CoV-2) pandemic has made it clear that combating coronavirus outbreaks benefits from a combination of vaccines and therapeutics. A promising drug target common to all coronaviruses—including SARS-CoV, MERS-CoV, and SARS-CoV-2—is the papain-like protease (PLpro). PLpro cleaves part of the viral replicase polyproteins into non-structural protein subunits, which are essential to the viral replication cycle. Additionally, PLpro can cleave both ubiquitin and the ubiquitin-like protein ISG15 from host cell substrates as a mechanism to evade innate immune responses during infection. These roles make PLpro an attractive antiviral drug target. Here we demonstrate that ubiquitin variants (UbVs) can be selected from a phage-displayed library and used to specifically and potently block SARS-CoV-2 PLpro activity. A crystal structure of SARS-CoV-2 PLpro in complex with a representative UbV reveals a dimeric UbV bound to PLpro at a site distal to the catalytic site. Yet, the UbV inhibits the essential cleavage activities of the protease *in vitro* and in cells, and it reduces viral replication in cell culture by almost five orders of magnitude.

## Author summary

In the last 3 years, it has become clear that emerging coronaviruses still pose a significant threat to human health. Despite the vaccines that are now available in many countries against severe acute respiratory syndrome coronavirus 2 (SARS-CoV-2), vaccine scarcity or the inability to gain proper protection from immunization due to an immunocompromised status still leaves various people that will benefit from the administration of potent

**Funding:** This work was supported by grants #2161 to GKA and SSS and #2189 to SSS from Emergent Ventures through the Thistledown Foundation (Canada) and the Mercatus Center at George Mason University and NIH grants (P01AI120943, R01AI123926 and R01AI161374 to GKA). This study was also supported by CIHR operating grants (COVID-19 Rapid Research Funding #OV3-170346 to SSS and BLM and, in part, #OV3-170649 to SSS). MK has received funding from the European Union's Horizon 2020 research and innovation program under grant agreement No 952373. The funders had no role in study design, data collection and analysis, decision to publish, or preparation of the manuscript.

**Competing interests:** The authors have declared that no competing interests exist.

and safe antiviral agents. In this research, we have developed ubiquitin variants (UbVs), small proteins that resemble a natural component of human cells called ubiquitin, that are able to bind a part of the virus that is essential for its survival, with high affinity and selectivity. This way, these UbVs are able to inhibit the production of new viral particles after infection, thus preventing the virus from spreading from cell to cell and wreaking havoc on the body.

## Introduction

Despite the rapid approval of effective vaccines, the COVID-19 pandemic caused by SARS-CoV-2 continues to affect human health across the globe more than 3 years after its emergence [1,2]. While initial vaccine scarcity and vaccination hesitancy by many have contributed to the continued circulation of the virus, breakthrough infections, waning immunity, and the emergence of new variants have also played a significant role in sustaining transmission of SARS-CoV-2 [3–6]. The reality is that SARS-CoV-2 is likely to become endemic in the human population and, unlike its predecessor SARS-CoV, human infections may never be eradicated completely. As we transition to co-existence with SARS-CoV-2, vulnerable populations—including the elderly and immunocompromised–may still endure poorer outcomes, irrespective of vaccine uptake [4]. Thus, it is critical to identify new antiviral therapies to help infected individuals to recover more quickly and to prevent life-threatening disease.

SARS-CoV-2 is an enveloped virus with a positive-sense, single-stranded RNA genome, and it belongs to the family of *Coronaviridae* within the order *Nidovirales* [7]. Like its relatives MERS-CoV and SARS-CoV, SARS-CoV-2 has a genome of approximately 30 kb and encodes various structural and accessory proteins in the 3' one-third of the genome. At least 16 non-structural proteins (nsps) are released from the polyproteins pp1a and pp1ab encoded in the 5' two-thirds of the genome. The processing of the polyproteins into functional nsps relies on two proteases, both part of the viral polyprotein itself: the papain-like protease (PLpro) in nsp3 and the 3C-like or main protease (3CLpro or Mpro) in nsp5. During infection, each enzyme cleaves at distinct sites of the polyproteins [8]. Whereas PLpro cleaves only three sites to release nsp1-3, its role in virus propagation is no less essential than that of Mpro, which cleaves all other sites [8–12]. Additionally, besides the essential role of PLpro in polyprotein cleavage, it possesses abilities that help the virus to evade the cellular antiviral innate immune response [13].

The innate immune system constitutes a first line of defense against incoming pathogens, including viruses, by inducing an antiviral state upon infection. Many viruses, however, have mechanisms in place to partially evade or delay the immune response. This impaired innate immunity has been associated with worse outcomes for COVID-19 patients [14]. Interestingly, PLpro of SARS-CoV-2, like those of SARS-CoV and MERS-CoV, has such an immune evasive effect, due to its deubiquitinating (DUB) activity [15]. This means that it is able to bind and cleave ubiquitin (Ub), covalently attached via isopeptide bonds, from substrates [13]. Ubiquitination of proteins is a post-translational modification (PTM) that has important roles in virtually all cellular processes and involves a highly regulated conjugation process directed by E1, E2 and E3 enzymes [16].

Ubiquitination is important in the innate immune response pathways as it often causes activation of the cellular factors necessary to produce an antiviral interferon response. Deubiquitination of these pathway components by the viral PLpro therefore inhibits the cellular immune response. For example, it was found that SARS-CoV PLpro can deubiquitinate signaling

molecules STING, TBK1, and IRF3, which suspended the cellular antiviral response [17]. Additionally, SARS-CoV-2 PLpro also has deISGylation activity, which removes the ubiquitin-like molecule interferon-stimulated gene 15 (ISG15) from host proteins [13]. The conjugation of ISG15 to cellular substrates evokes defensive antiviral effects via various routes [18,19]. In addition, it was found that ISGylation of the melanoma differentiation-associated protein 5 (MDA5) sensor is necessary for its activation, but that the protein is deISGylated by PLpro [20]. DeISGylation by PLpro is therefore another means whereby the virus evades the innate immune response. Consequently, drugs that inhibit the activity of PLpro will interfere in a myriad of processes required for virus propagation and innate immune evasion, and thus, are likely to be effective antiviral agents for the treatment of COVID-19.

The suitability of viral proteases as therapeutic targets has been established previously, as approved drug cocktails against both the human immunodeficiency virus (HIV) and hepatitis C virus (HCV) contain protease inhibitors [21–25]. Furthermore, Mpro of various coronaviruses has been the focus of numerous antiviral drug development programs, highlighting the recognition of viral proteases as promising drug targets [26,27].

Here, we have applied a unique protein engineering strategy to target the SARS-CoV-2 PLpro with methods that we previously applied to MERS-CoV PLpro [15]. We used phage-displayed libraries of billions of Ub variants (UbVs) to rapidly isolate and optimize UbV proteins that bind tightly and specifically to PLpro and inhibit polyprotein processing as well as the immune evasive functions of PLpro, both *in vitro* and in cells. Importantly, UbVs do not interfere with the ubiquitin (Ub) machinery of the cell, in this way avoiding adverse side effects. Most importantly, expression of one such UbV in cells strongly inhibited the replication of SARS-CoV-2, thus validating PLpro as a target for antiviral therapies and confirming the innovative concept of using UbVs as antiviral agents. Unexpectedly, the crystal structure of a UbV in complex with SARS-CoV-2 PLpro showed that the UbV does not bind PLpro in the same way as natural Ub, but instead, binds as a strand-swapped dimer far away from the active site, thus revealing an alternative inhibitory mechanism. Taken together, these results uncover a novel mechanism of SARS-CoV-2 PLpro inhibition that could provide new means for the development of COVID-19 therapies.

## Results

### Engineered UbVs bind to SARS-CoV-2 PLpro

We previously used phage display to engineer UbVs that inhibited the activity of the MERS-CoV PLpro domain [15], and here, we applied the same strategy to SARS-CoV-2 PLpro. The SARS-CoV-2 PLpro enzyme was expressed as a soluble protein in *Escherichia coli* with its N-terminus fused to a His$_6$-GST affinity tag (S1 Fig). The ~60 kDa fusion protein was stable and could be cleaved efficiently into ~25 kDa GST and ~35 kDa PLpro using HRV-3C PreScission protease. The His$_6$-GST-PLpro fusion protein was used as a target for five rounds of binding selections with a phage-displayed library of UbVs. Phage ELISAs were used to identify individual positive clones that bound to PLpro but not to GST, bovine serum albumin (BSA), or streptavidin (SAV). We subjected 96 positive clones to DNA sequencing, but all clones were found to be identical in sequence (UbV.CV2.1, Fig 1A).

To further optimize binding, we constructed a second-generation library in which the sequence of UbV.CV2.1 was subjected to a soft randomization strategy in which a set of residues were diversified such that approximately 50% of the sequence at each position was the original UbV.CV2.1 sequence and 50% was randomly mutated [28]. Following five rounds of selection for binding to PLpro, DNA sequencing of 96 positive clones yielded 10 unique sequences (UbV.CV2.1a-j) that contained 13–18 substitutions relative to Ub.wt (Fig 1A).

**A**

| | Region 1 | | | | | | | Region 2 | | | | Region 3 | | | | | | | | | | | Kd (nM) | IC$_{50}$ (nM) |
|---|---|---|---|---|---|---|---|---|---|---|---|---|---|---|---|---|---|---|---|---|---|---|---|---|
| | 2 | 4 | 9 | 10 | 11 | 12 | 14 | 44 | 46 | 48 | 49 | 62 | 63 | 64 | 68 | 70 | 72 | 74 | 75 | 76 | 77 | 78 | | |
| **Ub WT** | Q | F | T | G | K | T | T | I | A | K | Q | Q | K | E | H | V | R | R | G | G | - | - | | |
| UbV.CV2.1 | - | S | M | R | - | R | - | - | S | M | L | - | - | G | T | G | I | - | A | N | G | V | 14.0 | 11.8 |
| UbV.CV2.1a | - | T | L | R | M | R | - | L | - | M | L | - | - | G | T | G | V | P | S | S | G | V | 1.8 | 5.5 |
| UbV.CV2.1b | E | T | L | R | M | K | A | - | R | L | L | - | - | G | T | G | V | - | D | N | A | V | 1.0 | 1.8 |
| UbV.CV2.1c | - | S | M | R | - | W | - | - | Y | M | H | E | - | G | T | G | I | G | L | P | G | V | n/a | n/a |
| UbV.CV2.1d | - | Y | L | R | - | R | - | - | H | M | L | - | - | G | T | G | I | T | S | S | G | V | n/a | n/a |
| UbV.CV2.1e | - | T | L | R | - | R | - | - | G | - | L | - | - | G | T | G | I | G | A | N | G | V | n/a | n/a |
| UbV.CV2.1f | - | A | L | R | - | - | N | - | S | - | - | - | - | G | T | G | I | G | I | S | G | V | 1.5 | 8.3 |
| UbV.CV2.1g | - | S | V | R | R | - | I | L | - | M | L | R | - | G | T | G | I | G | A | S | G | V | 1.0 | 8.7 |
| UbV.CV2.1h | R | L | L | R | - | M | S | - | Y | T | L | E | T | G | T | G | I | H | S | A | G | G | 7.0 | 10.3 |
| UbV.CV2.1i | - | S | M | R | - | R | - | V | H | M | L | - | R | D | T | G | I | S | A | H | S | V | 4.0 | 11.0 |
| UbV.CV2.1j | - | S | L | R | - | - | - | N | F | M | L | - | - | G | T | G | V | G | A | N | Y | V | 1.6 | 12.6 |

**B**

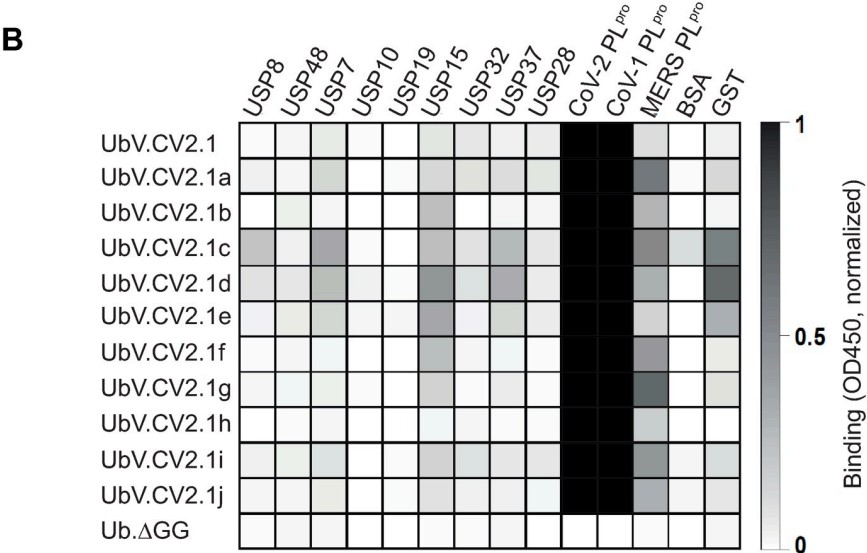

**Fig 1. Engineered UbVs selected for binding to SARS-CoV-2 PLpro. (A)** Sequence alignment of wt ubiquitin (Ub.wt) and UbVs selected for binding to SARS-CoV-2 PLpro. The residue numbers are shown at the top, and the alignment shows only those positions that were subjected to diversification in the phage-displayed UbV library. The dissociation constant (K$_D$) was measured using BLI with immobilized His$_6$-GST-PLpro and solution-phase His-FLAG-UbVs. The inhibitory concentration 50 (IC$_{50}$) was calculated as the concentration of UbV required to cause 50% inhibition of hydrolysis of ISG15-AMC by PLpro, **(B)** The binding specificities of FLAG-tagged UbVs (*y-axis*) are shown across a group of 12 DUBs (*x-axis*), including 9 different human DUBs (USP7, USP8, USP10, USP15, USP19, USP28, USP31, USP37, and USP48) (x-axis), viral PLpro enzymes (SARS-CoV, MERS-CoV, and SARS-CoV-2), and two negative controls (GST and BSA), as assessed by protein ELISA. Purified His$_6$-FLAG-UbVs were added to immobilized proteins as indicated and bound UbVs were detected by the addition of anti-FLAG-HRP and colorimetric development of TMB peroxidase substrate. The mean value of absorbance at 450 nm is shaded in a grayscale.

Notably, a subset of these substitutions were completely or highly conserved amongst the variants, but differed from Ub.wt (T9L, G10R, K48M, Q49L, E64G, H68T, V70G, R72I), suggesting that these substitutions are likely responsible for the enhanced affinity.

The UbVs were expressed in *E. coli* using a construct that also encodes an N-terminal hexa-His tag followed by a FLAG tag, and the fusion proteins were purified by NiNTA affinity chromatography. We assessed the specificities of the UbVs by ELISA with immobilized target proteins and detection of bound UbV protein with an anti-FLAG antibody (**Fig 1B**). ELISAs were conducted with a panel of proteases, including the PLpro enzymes of SARS-CoV-2,

SARS-CoV and MERS-CoV, and seven human USP enzymes that belong to the same structural class as the PLpro enzymes. None of the UbVs bound to BSA but three (UbV.CV2.1c, d, e) bound to GST, suggesting non-specific interactions, so these three were left out from further analysis. The remaining UbVs all bound strongly to the SARS-CoV-2 and SARS-CoV PLpro enzymes and most bound moderately to the MERS-CoV PLpro, but they all exhibited minimal or no binding to the human USPs, decreasing the likelihood of adverse binding of virus-specific UbVs to cellular enzymes that have structural similarity to PLpro.

To accurately measure the affinities of the UbVs for SARS-CoV-2 PLpro, we used biolayer interferometry (BLI) to determine dissociation constants for immobilized PLpro and UbVs in solution (**Figs 1A and S2**). The parent UbV.CV2.1 bound with high affinity ($K_D$ = 14 nM) and all the optimized UbVs bound with enhanced affinities ($K_D$ = 1.0–7.0 nM). We also assessed inhibition of deISGylation by UbVs with an *in vitro* assay that measured the hydrolysis of the fluorogenic substrate ISG15-AMC by purified PLpro. This assay demonstrated that UbV. CV2.1 and its optimized variants potently inhibited the hydrolysis of ISG15-AMC ($IC_{50}$ = 1.8– 13 nM, **S3 Fig**). Taken together, these results showed that the phage display technology was successful for selecting and engineering UbVs that bound tightly and specifically to SARS-CoV-2 PLpro and inhibited its deISGylation activity.

## UbVs inhibit the deISGylation and deubiquitination activities of SARS-CoV-2 PLpro

We next sought to determine whether the UbVs could inhibit the catalytic activities of SARS-CoV-2 PLpro in cell-based assays. PLpro has been shown to have strong proteolytic activity for cleavage of ISG15 chains, dubbed deISGylation [13]. We performed cell-based deISGylation assays to assess whether the UbVs could inhibit this activity. Plasmids encoding SARS-CoV-2 PLpro and the UbVs were co-transfected with the different components necessary for ISGylation of cellular substrates. Unlike the ubiquitination of proteins, which can be visualized by simply transfecting Ub.wt into cells, the enzymes needed for ISGylation (E1 activating enzyme UbE1L, E2 conjugating enzyme UbcH8 and E3 ligase HERC5) are normally only induced by type I interferon and were therefore co-expressed to achieve ISGylation in this assay [18]. Upon co-expression of all these components with the plasmid that encodes for SARS-CoV-2 PLpro, ISGylation of cellular substrates was almost completely abrogated (**Fig 2**, compare lanes 2 and 3). This was not the case when the catalytic mutant of PLpro ('C') was co-transfected instead (lane 4). After co-expression of wildtype PLpro with the various UbVs, it was clear that UbV.CV2.1a, b, and i inhibited the deISGylating activity of PLpro and partially restored the ISGylation of cellular substrates (**Fig 2**, compare lane 3 to lanes 5–12). A construct that expresses a mutated form of ubiquitin (Ub.AA, which lacks a free di-glycine motif and prevents their conjugation to cellular substrates) was used as a negative control and had no effect on PLpro deISGylating activity. We similarly examined the effect of the UbVs on the DUB activity of SARS-CoV-2 PLpro. As expected, SARS-CoV-2 PLpro only had modest deubiquitinating activity (**S4 Fig**). Nonetheless, UbVs 1a, b, h and j were able to inhibit the DUB function of PLpro.

## UbVs inhibit the polyprotein cleavage activity of SARS-CoV-2 PLpro

The ability of PLpro to cleave the viral replicase polyproteins at three sites to release nsp1, nsp2 and nsp3 is essential for viral replication [29,30]. Thus, we tested the effects of the UbVs on polyprotein cleavage *in vitro* (**Fig 3A**). As substrate, we used a polyprotein fragment spanning an N-terminal part of nsp3 (nsp3N), also containing the active PLpro domain, as well as a C-terminal part of nsp2 (nsp2C), between which PLpro is known to cleave (**S5A Fig**). As

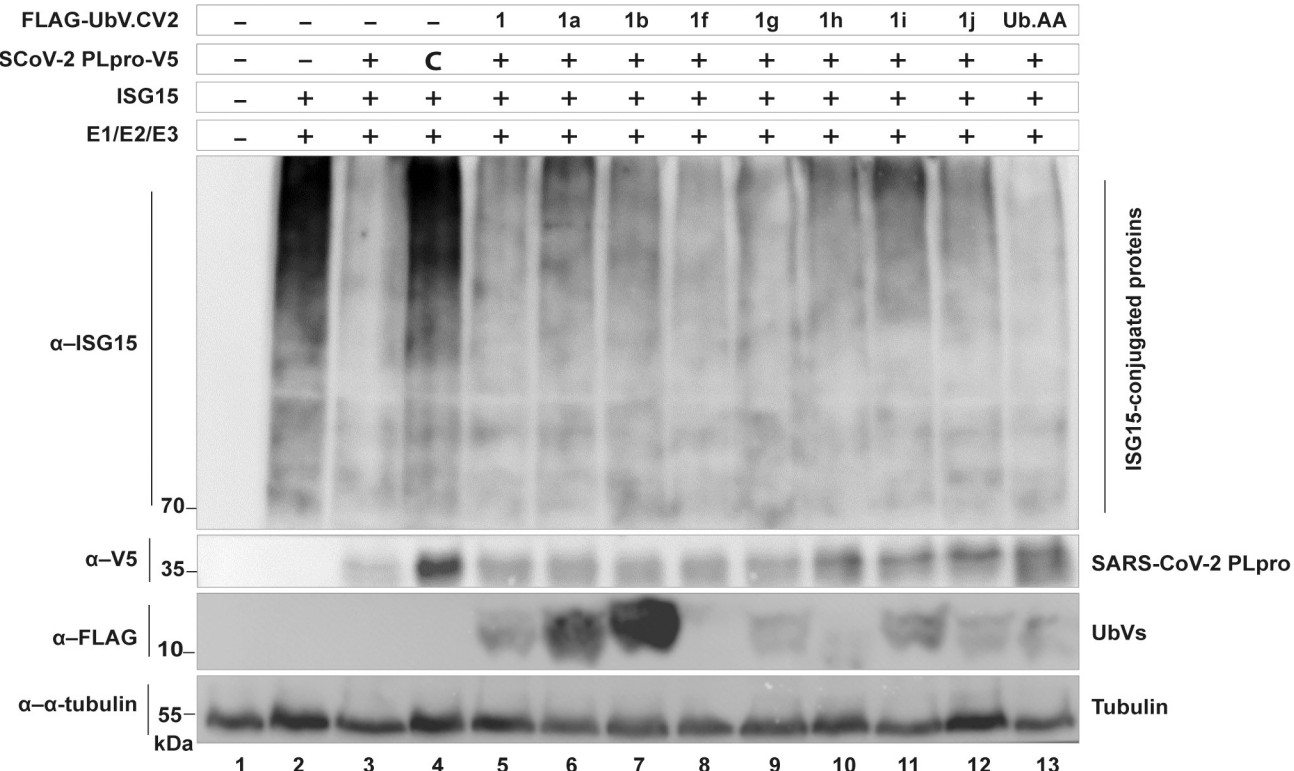

**Fig 2. UbVs inhibit the deISGylation activity of SARS-CoV-2 PLpro.** Inhibition of the deISGylation activity of SARS-CoV-2 PLpro by some UbVs, visualised by co-transfection of HEK293T cells with plasmids encoding for the E1, E2, and E3 enzymes necessary for conjugation of ISG15, together with hISG15, as well as SARS-CoV-2 PLpro-V5 (wildtype or catalytic mutant 'C') and FLAG-UbVs. Lysates were collected 48 hours post transfection and were subjected to western blot analysis. Whereas PLpro normally removes ISG15 from cellular substrates, co-transfection of some the UbVs inhibits the deISGylation activity of PLpro.

expected, *in vitro* transcribed and translated V5-nsp2C-nsp3N-HA was quickly processed into two smaller fragments while the inactive catalytic (C111A) PLpro mutant remained as a single, large polypeptide (**Fig 3A**, compare lanes 2 and 3) [8,31]. Addition of purified UbV.CV2.1a or UbV.CV2.1b effectively blocked proteolytic processing in a dose-dependent manner (**Fig 3A**, compare lane 2 to lanes 6–11), whereas a wt Ub lacking the C-terminal di-Gly motif (Ub. ΔGG) had no effect even at the highest concentration (**Fig 3A**, lane 5). Thus, UbV.CV2.1a and 1b potently inhibited the nsp2↓nsp3 cleavage by PLpro *in vitro*. Similar experiments were conducted using constructs encoding for the PLpro domain of the alpha and beta variants of SARS-CoV-2. UbVs 1a and 1b efficiently blocked proteolytic activities of variant-derived PLpro enzymes (**S6 Fig**).

We also confirmed the inhibitory properties of two high-affinity UbVs in a cellular model of polyprotein cleavage by PLpro. We transiently transfected HEK293 cells with an expression plasmid encoding the PLpro domain together with a plasmid encoding a fragment that encodes the C-terminal part of nsp3, connected to the N-terminal part of nsp4 (**S5B Fig**). PLpro could cleave this nsp3C-nsp4N fragment, and cleavage was dependent on its catalytic cysteine [15] (**Fig 3B**). Co-transfection with plasmids encoding UbV.CV2.1a or UbV.CV2.1b resulted in strong inhibition of polyprotein cleavage in a dose-dependent manner, as evidenced by the presence of the higher molecular weight V5-nsp3C-nsp4N-HA substrate, which remained uncleaved. In contrast, co-transfection with a plasmid encoding wt Ub with a di-Ala C-terminal tail (Ub.AA) had no effect on polyprotein cleavage.

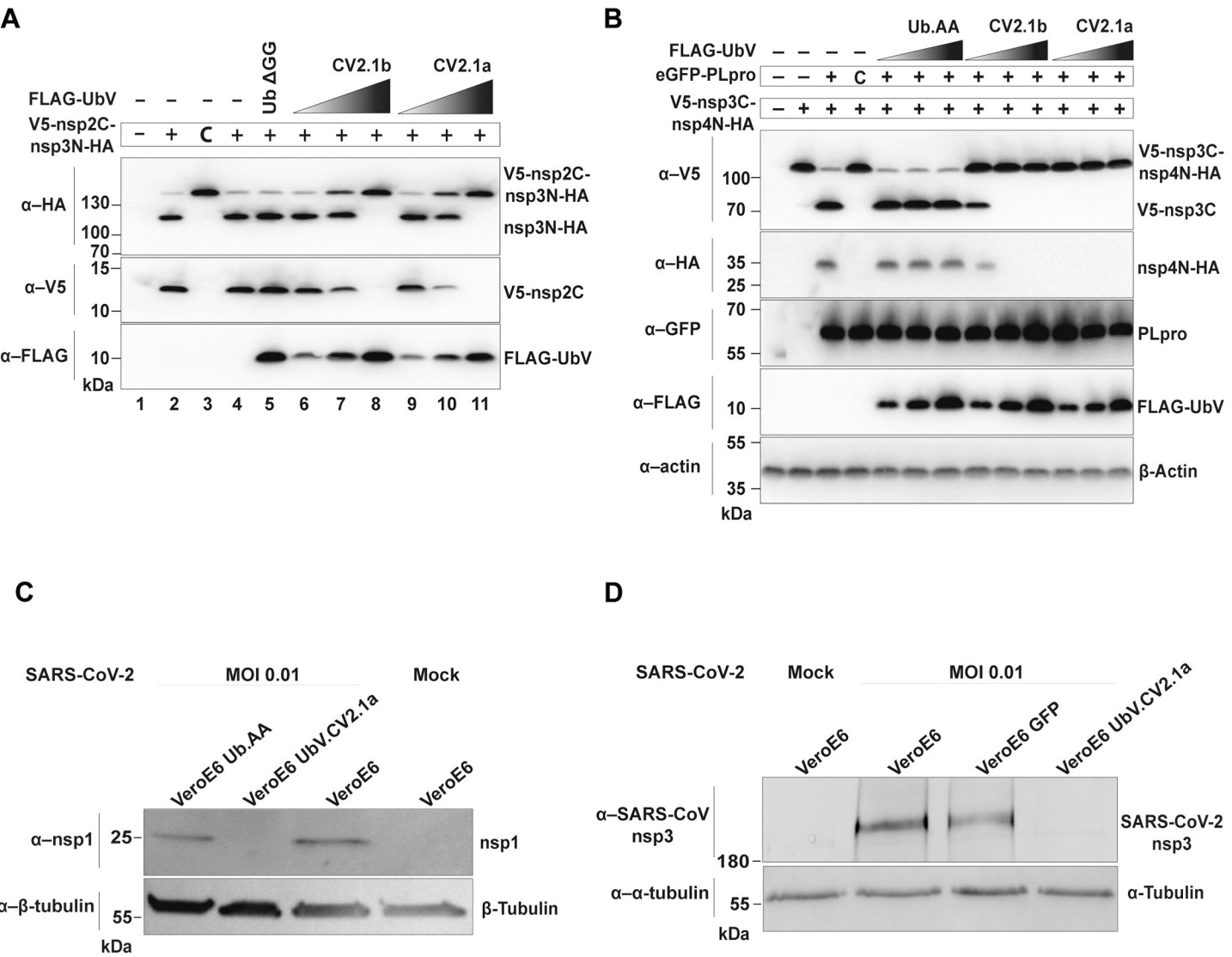

**Fig 3. UbVs inhibit proteolytic activity of SARS-CoV-2 PLpro *in vitro* and in cell culture. (A)** The ability of UbV.CV2.1a and 1b to inhibit proteolytic activity of SARS-CoV-2 PLpro was assessed *in vitro* at increasing concentrations. A construct encoding V5-nsp2C-nsp3N-HA—N-terminal V5-tagged and C-terminal HA-tagged nsp2C-nsp3N (including the PLpro domain)—was transcribed and translated *in vitro* in the presence of 0.1, 1.0, or 10 μM UbV.CV2.1a or 1b for 2 hours at 37˚C. Western blotting using the indicated antibodies was used to detect the presence of N-terminal V5-tagged nsp2C and C-terminal HA-tagged nsp3N cleavage products. **(B)** Inhibition of proteolytic activity of SARS-CoV-2 PLpro by UbVs was examined in transiently transfected HEK293T cells. N-terminal V5-tagged and C-terminal HA-tagged nsp3C-nsp4N (V5-nsp3C-nsp4N-HA) that excludes the PLpro domain was co-expressed with GFP-tagged SARS-CoV-2 PLpro (wt or the catalytically inactive C111A mutant 'C') and FLAG-UbV.CV2.1a or 1b (with increasing amounts of 0.5, 0.75, and 1.0 ug). Cells were lysed 20 hours post transfection and analyzed by western blotting with the indicated antibodies to detect generation of N-terminal V5-tagged nsp3C and C-terminal HA-tagged nsp4N cleavage products. **(C)** Effect of UbV. CV2.1a on SARS-CoV-2 mature nsp1 production. Representative western blot analysis of nsp1 in VeroE6 cells stably expressing empty vector (VeroE6), Ub.AA or UbV. CV2.1a infected with SARS-CoV-2 at a MOI of 0.01 and harvested 48 hpi. Uninfected (mock) parental VeroE6 cells were included as a control. β-tubulin served as a loading control. **(D)** Effect of UbV.CV2.1a on SARS-CoV-2 nsp3 production and processing. Western blot analysis was performed using the indicated antibodies of the same cells as (C) infected with SARS-CoV-2 at an MOI of 0.01 and lysed 48 hpi. α-Tubulin served as a loading control.

We then examined whether this effect of UbV.CV2.1a could be confirmed in cell-based assays upon SARS-CoV-2 infection (**Fig 3C**). VeroE6 cells stably expressing empty vector (EV), a FLAG-tagged wt Ub with a di-Ala C-terminal tail (Ub.AA), or FLAG-UbV.CV2.1a, were infected with SARS-CoV-2 at a multiplicity of infection (MOI) of 0.01. Two days after infection, cells were lysed and subjected to western blot analysis to monitor maturation, and thus, cleavage of the polyprotein. Presence of the mature and cleaved non-structural protein 1

(nsp1) can be seen in infected VeroE6 cells, as well as in control cells expressing Ub.AA. However, nsp1 is absent in cells expressing UbV.CV2.1a (**Fig 3C**). A similar result was also found upon detection of nsp3 in infected VeroE6 cells, but not in the cells expressing the UbV (**Fig 3D**), suggesting inability of the virus to produce mature viral proteins in the presence of the UbV.

## UbV.CV2.1a strongly inhibits SARS-CoV-2 replication

To assess whether UbV.CV2.1a interferes with SARS-CoV-2 replication in infected cells, VeroE6 cells stably expressing controls (empty vector (EV) or Ub.AA) or FLAG-UbV.CV2.1a were infected with SARS-CoV-2 at a multiplicity of infection (MOI) of 0.01, after which viral load in the culture supernatant was measured by quantitative RT-PCR using primers specific for the Nucleocapsid (N) gene. At 24 hours post-infection (hpi), the amount of viral RNA produced by cells expressing UbV.CV2.1a was significantly reduced (**Fig 4A**), which was also clear after re-infection of wt VeroE6 cells with harvested supernatants (**Fig 4B**).

Furthermore, this finding was confirmed by studying the production of infectious virions. Parental VeroE6 cells, as well as those stably expressing FLAG-tagged UbV.CV2.1a or GFP, were infected with SARS-CoV-2 at an MOI of 0.01 or 1. Supernatants and lysates were harvested at both 24 and 48 hpi, and wells containing cells on coverslips were fixed in 3% paraformaldehyde (PFA). The supernatants were subjected to titration by plaque assay. Following infection at an MOI of 0.01, the VeroE6 parental cells yielded a titre of $\sim 1.5 \times 10^4$ plaque forming units (PFU)/mL, whereas from the UbV.CV2.1a-expressing VeroE6 cells, only $\sim 40$ PFU/mL could be recovered at 24 hpi (**Fig 4C**). After 48 hpi, the viral titre from the cells expressing UbV.CV2.1a remained at only $\sim 20$ PFU/mL, whereas the titre from the parental cells increased to $1 \times 10^6$ PFU/mL. The control cells expressing GFP showed comparable viral titres to the parental cells. Thus, expression of UbV.CV2.1a reduced infectious SARS-CoV-2 titres by up to almost 5 orders of magnitude. A similar trend was observed when the cells were infected at an MOI of 1 (**Fig 4D**). At 24 hpi, SARS-CoV-2 infection of cells expressing UbV.CV2.1a resulted in substantially lower viral titres ($6 \times 10^3$ PFU/mL) compared to the parental cells ($8 \times 10^5$ PFU/mL) and the cells expressing GFP. At 48 hpi, parental cells produced a viral titre of $2 \times 10^6$ PFU/mL, whilst the cells expressing UbV.CV2.1a produced only $5 \times 10^3$ PFU/mL. Thus, following a SARS-CoV-2 infection with a MOI of 1, viral titres in cells expressing UbV. CV2.1a were decreased by at least 2 orders of magnitude at both 24 and 48 hpi, compared to control cells.

These results were verified by western blot analysis of the harvested lysates (**Fig 4E**). Western blot membranes were stained for SARS-CoV-2 M protein, besides the appropriate controls for the presence of GFP, the presence of the UbV.CV2.1a (using an anti-FLAG antibody), and alpha-tubulin as a loading control. In lysates of cells harvested at 24 hpi with an MOI of 0.01, too little M protein was present to detect in all cells, but in lysates taken 48 hpi, M protein was only detected in the parental cells and the cells expressing GFP, but not in the cells expressing UbV.CV2.1a. Similar results were found for the cells infected with an MOI of 1, where both 24 and 48 hpi, no M protein was detected in the cells expressing UbV.CV2.1a, whereas the protein was clearly present in parental cells and cells expressing GFP.

Furthermore, the same trend was observed by immunofluorescence microscopy (**Fig 4F**). On coverslips, parental cells and cells expressing UbV.CV2.1a were infected with SARS-CoV-2 at an MOI of 0.01 and fixed in PFA 48 hpi. The cells were then stained for SARS-CoV-2 M protein, as well as the FLAG-UbV using an anti-FLAG antibody. Parental cells showed a strong positive signal for the viral M protein. The cells expressing UbV.CV2.1a showed a strong signal for FLAG-UbV, but the signal for M protein was drastically reduced and was

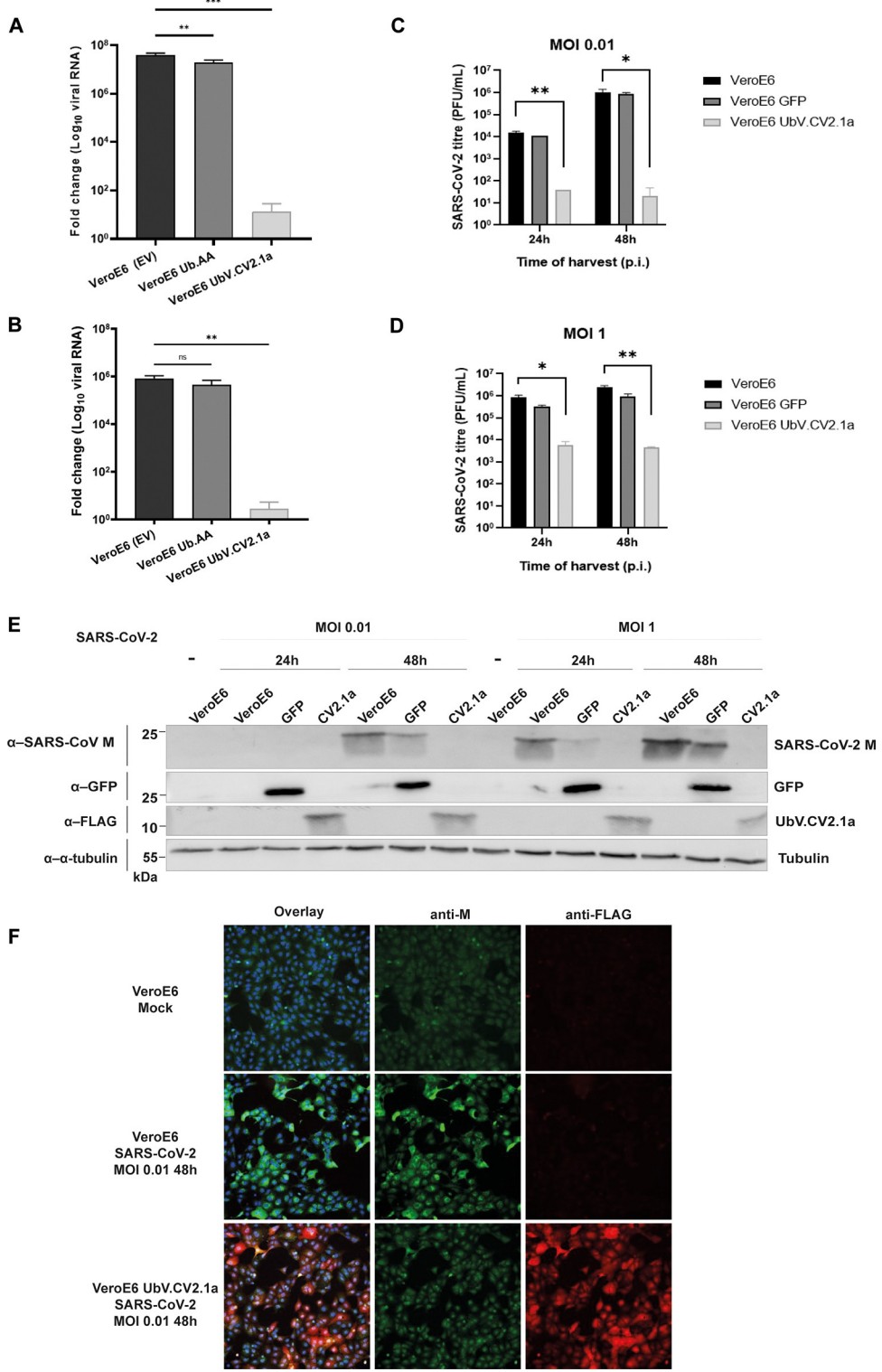

**Fig 4. UbV inhibits SARS-CoV-2 replication.** (**A**) SARS-CoV-2 replication was assessed 24 hpi in the presence or absence of UbV.CV2.1a by quantitative RT-PCR using primers specific for the viral N-gene. Stable VeroE6 cells expressing empty vector (black bars) or Ub.AA (dark grey bars) served as controls. UbV.CV2.1a cells are shown in white bars. Data are normalized to β-actin control. (**B**) A viral propagation assay was performed by using supernatants collected from SARS-CoV-2 infected stable VeroE6 cells to re-infect parental VeroE6 cells and the levels of N gene expression were measured by quantitative RT-PCR as in (A). (**C and D**) Viral replication was also assessed by

measuring plaque forming units (PFUs) of virus that were released into the supernatant of the indicated parental (black bars) or stable (grey bars for GFP and white bars for UbV.CV2.1a) cells. Cells were infected with an MOI of 0.01 (**C**) or an MOI of 1 (**D**) and SARS-CoV-2 titres were subsequently determined for supernatants harvested at 24 and 48 hpi. (**E**) Western blot analysis of lysates from (**C**) and (**D**) using antibodies that recognize SARS-CoV-2 M protein, GFP, FLAG, or α-tubulin. (**F**) Immunofluorescence was performed to examine the level of SARS-CoV-2 M protein expression in parental and UbV.CV2.1a-expressing VeroE6 cells infected with SARS-CoV-2 at MOI 0.01 and fixed at 48 hpi. M protein is shown in green, FLAG-UbV is shown in red, and Hoechst is in blue. Mean ± SD, ns = non-significant, $^*$ $p < 0.05$, $^{**}$ $p < 0.01$, and $^{***}$ $p < 0.001$.

comparable to the background signal observed in mock-infected cells. Taken together, these results showed that—in cells infected with SARS-CoV-2—the production of viral RNA, M protein, and infectious progeny virus was greatly reduced upon expression of UbV.CV2.1a, again confirming the antiviral effect of this UbV.

## Structural basis for inhibition of SARS-CoV-2 PLpro by UbV.CV2.1

To better understand the mechanism underlying inhibition, we determined the crystal structure of SARS-CoV-2 PLpro in complex with UbV.CV2.1 at a resolution of 3.5 Å (**Table 1**). The complex revealed that UbV.CV2.1 bound to PLpro as an asymmetric strand-swapped dimer (**Figs 5A and S7**). This conformation was unexpected, since it was not observed previously in the structure of UbV.ME.4 bound to MERS-CoV PLpro [15], and it revealed key differences in the way UbV.CV2.1 engaged its target. Structural superposition showed no overlap between the monomeric UbV.ME.4 inhibitor bound to MERS-CoV PLpro, which bound close to the catalytic site, and the dimeric UbV.CV2.1 bound to SARS-CoV-2 PLpro, which bound to a site far from the catalytic site (**Fig 5B**). However, there was overlap between the N-terminal domain of ISG15 bound to SARS-CoV-2 PLpro and one half of the UbV.CV2.1 dimer (**Fig 5B**), suggesting that UbV.CV2.1 inhibits ISGylation by sterically blocking the interaction between PLpro and the N-terminal domain of ISG15.

The contacts between SARS-CoV-2 PLpro and UbV.CV2.1 result in an extensive binding interface with 1063 and 1040 Å$^2$ of surface area buried on the protease and UbV, respectively (**Fig 5C**). The binding site on PLpro can be divided roughly into two halves, which we have termed subsite-1 (412 Å$^2$ that predominantly contacts UbV) and subsite-2 (651 Å$^2$ that predominantly contacts UbV').

Subsite-1 involves hydrophobic burial of one face of the PLpro S2 helix (residues 60–70$^*$, PLpro residues are denoted by asterisks throughout) by the UbV subunit (**Fig 5D**), analogous to interactions of the N-terminal domain of ISG15 with PLpro. UbV residues Ile44 and Thr68 bury PLpro residue Thr63$^*$; residues Arg42, Leu73 and Leu8' (from the UbV' subunit) interact with residues Val66$^*$ and Phe69$^*$; and Leu8 serves to bury residue Glu70$^*$. Hydrophilic interactions also occur, as the backbone amides of Ser46 and Gly47 make hydrogen bonds with the sidechain of Asp62$^*$. Three UbV residues that differ from Ub.wt stand out in subsite-1. It has been shown that substitutions to more bulky residues in place of the β1-β2 loop residues Thr9 and Gly10 favor formation of swapped dimers, and in UbV.CV2.1, these residues correspond to Met9 and Arg10. Met9' points toward the solvent, but Arg10' from UbV' makes a salt bridge with Glu70$^*$ of PLpro. The third residue of interest is at position 68, where Thr68 in the UbV is accommodated at the interface, but His68 of Ub.wt would cause a steric clash.

Subsite-2 involves interactions with two distinct regions of PLpro. In the first region, residues from strand β4 of the UbV' subunit interact with strand β2 of the Ubl domain of PLpro (residues 14–18$^*$) to form a continuation of the β-sheet, with 4 mainchain hydrogen bonds between the two strands (**Fig 5E,** left). This interaction results in the hydrophobic burial of UbV' residues Gly70' and Ile72' by PLpro residues Phe8$^*$, Ile14$^*$ and Leu16$^*$ on one side of the

**Table 1. Data Collection and Refinement Statistics for the X-ray Structure of the SARS-CoV-2 PLpro/UbV.CV2.1 Complex.**

| *Data Collection* | |
|---|---|
| X-ray source | Rigaku R-AXIS IV++ |
| Wavelength (Å) | 1.54178 |
| Crystal | native |
| Space group | P2$_1$ |
| Cell dimensions a, b, c (Å), β (˚) | 54.6, 174.9, 121.6, 95.7 |
| Resolution (Å) | 49.74–3.50 (3.71–3.50)[a] |
| Total no. unique reflections | 23394 (2882) |
| Mean [(I) /σ(I)] | 6.4 (2.5) |
| Completeness (%) | 81.9 (63.0) |
| R$_{merge}$ | 0.124 (0.377) |
| Multiplicity | 3.3 (3.4) |
| Wilson *B*-Factor | 61.47 |
| *Refinement Statistics* | |
| Resolution (Å) | 47.79–3.50 |
| No. of reflections | 23340 (1998*)[b] |
| R$_{work}$ / R$_{free}$ | 0.225/0.270 |
| RMSD bond lengths (Å) | 0.005 |
| RMDS bond angles (˚) | 0.95 |
| No. of protein atoms | 13973 |
| No. of ligand/ion atoms | 13 |
| Average B-factor protein | 63 |
| Average B-factor ligand/ion | 75 |
| *Ramachandran statistics (molprobity)* | |
| Preferred (%) | 92.1 |
| Allowed (%) | 7.9 |
| Disallowed | 0 |
| Clashscore | 17.1 |

[a]Values in parentheses represent the highest-resolution shell
[b]Number of reflections in the Rfree set.

β-sheet, and residues Leu71' and Leu73' by residues Asn15* and His17* on the other side. There is also a hydrogen bond between the sidechains of Arg42' and Asn13*. In this region, two residues that are substituted relative to Ub.wt are of interest. First, Gly70' comes very close (within 3 Å) to Ile14* of PLpro, whereas a Val70' sidechain of Ub.wt would cause a steric clash. Second, Ile72' fits in a hydrophobic pocket between residues Leu16* and Phe8* of PLpro, whereas the larger Arg72' of Ub.wt would be a less complementary fit. In the second region of subsite-2, residues Asp134*, Tyr137*, Arg138* and Ala141* in an α-helix of PLpro interact with residues Ser46', Gly47' and Met48' in a loop of UbV' (**Fig 5E**, right). This loop is held in position by burial of Met48' and Leu49' through intramolecular interactions within UbV', with Met48' making hydrophobic interactions with Tyr59', and Leu49' interacting with Arg42', Ile44' and Tyr59'. Notably, at positions 48 and 49, UbV' contains hydrophobic substitutions (K48M, Q49L) relative to Ub.wt.

To dissect subsites-1 and -2 on PLpro for their role in mediating the binding and inhibitory functions of UbV.CV2.1, we constructed PLpro variants containing mutations designed to

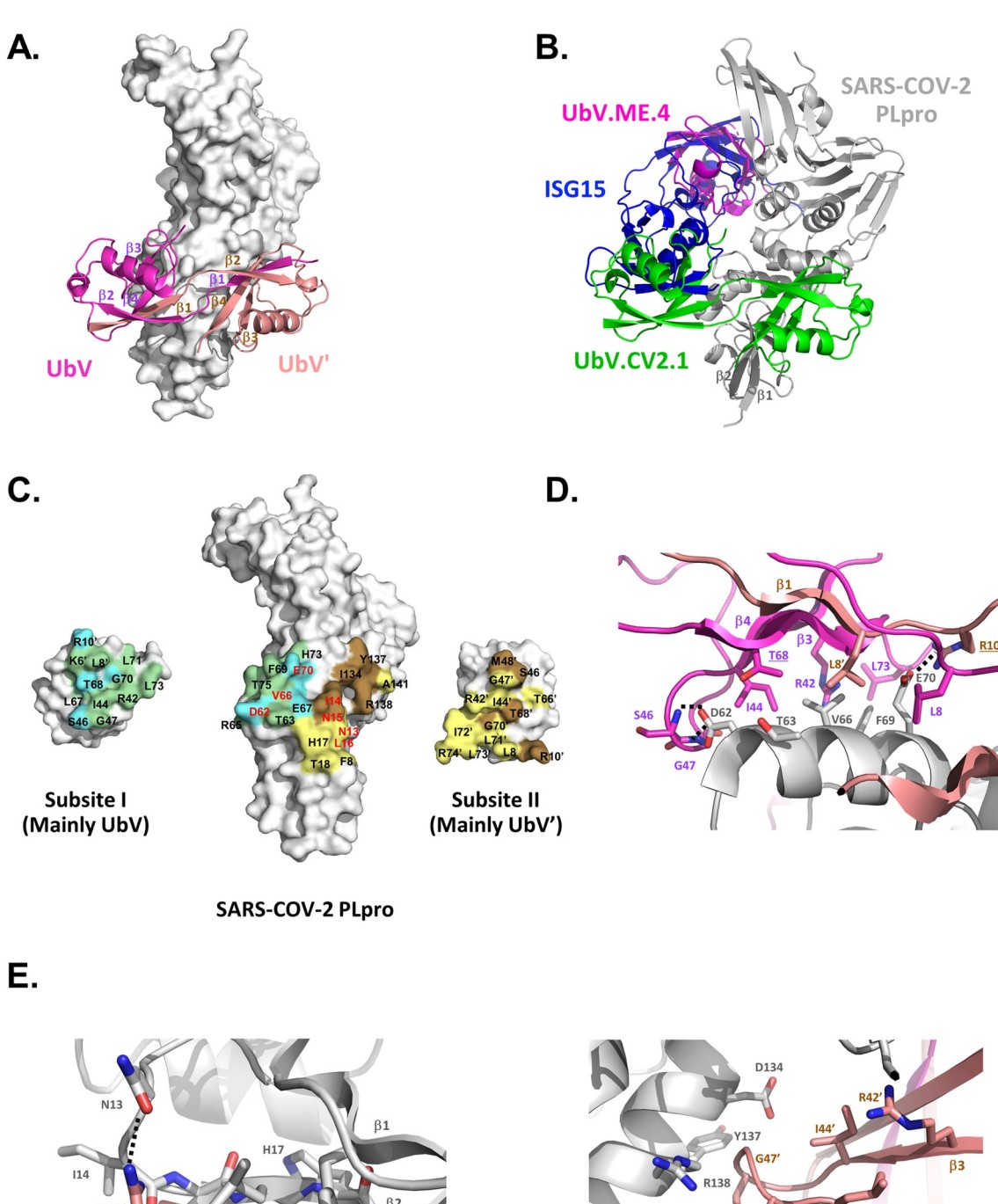

**Fig 5. Structure of the SARS-CoV-2 PLpro bound to a UbV.CV2.1. dimer.** . **(A)** Overall structure of the complex. PLpro is represented as a white molecular surface, and the UbV.CV2.1 dimer is represented as a ribbon with the promoters colored magenta (UbV) or salmon (UbV'). **(B)** Superposition of the SARS-CoV-2 PLpro (grey) in complex with the UbV.CV2.1 dimer (green) with the structure of ISG15 (blue) from the complex with the SARS-CoV-2 PLpro (PDB entry 6YVA), and the structure of UbV.ME4 monomer (magenta) from the complex with the MERS PLpro (PDB entry 5V69). **(C)** Open-book view of the complex. SARS-COV-2 PLpro is in

the center. The half of the UbV.CV2.1 dimer that interacts with subsite-1 (UbV residues 10–74 and UbV' residues 1–9) or subsite-2 (UbV' residues 10–74 and UbV residues 1–9) is at the left or right, respectively. PLpro and UbV.CV2.1 residues that are buried at the interface are colored green or blue (subsite-1) or yellow or brown (subsite-2) if they are the same or different from Ub.wt, respectively. Residues that were substituted with alanine in PLpro-S1 or PLpro-S2 are labelled in red. **(D-E)** Close-up views of the interactions between UbV.CV2.1 and PLpro **(D)** subsite-1 and **(E)** subsite-2, utilizing the color scheme in **(A)**. For clarity, beta strands of each UbV are labelled in parts **(A)**, **(C)** and **(D)**, and strands β1 and β2 of the SARS-CoV-2 PLpro involved in interactions in subsite II are labelled in parts **(B)** and **(D)**.

disrupt interactions with UbV.CV2.1 in either or both sites. PLpro-S1 and PLpro-S2 contained Ala substitutions in place of 3 or 4 residues in the center of subsite-1 (Asp62, Val66, Glu70) or subsite-2 (Asn13, Ile14, Asn15, Leu16), respectively, and PLpro-S1/2 contained substitutions in both subsites (**Fig 5C**). We first tested the effects of the substitutions on the binding interaction with UbV.CV2.1 using an *in vitro* pulldown assay with FLAG-tagged UbV and GFP-tagged PLpro proteins. Immunoprecipitation with anti-GFP antibody followed by immunoblotting with anti-FLAG antibody showed that, relative to PLpro-wt, binding of PLpro-S1 and PLpro-S2 to UbV.CV2.1 was significantly reduced, and the interaction with PLpro-S1/S2 was completely abrogated (**Fig 6A**).

Next, we assessed the effects of the PLpro substitutions on the inhibitory activity of UbV.CV2.1 in an *in vitro* assay of polyprotein processing using the V5-nsp2C-nsp3N-HA substrate described prior. Based on the lack of the full-length V5-nsp2C-nsp3N-HA band and the appearance of bands representing the cleavage products V5-nsp2C and nsp3N-HA, the proteolytic activity of PLpro-S2 was similar to that of PLpro-wt, whereas the activities of PLpro-S1 and PLpro-S1/S2 were significantly reduced, and as expected, the activity of the C111A catalytic mutant (C) was completely abrogated (**Fig 6B**) [8,31]. The addition of UbV.CV2.1 inhibited the polyprotein processing activity of PLpro-wt and PLpro-S2, but not that of PLpro-S1 and PLpro-S1/2 (**Fig 6C**).

Taken together, these results showed that subsite-1 and subsite-2 both contribute to binding of PLpro to UbV.CV2.1, as mutations in either subsite reduced binding and mutations in both subsites abrogated binding (**Fig 6A**). Most importantly, mutations in subsite-1 abrogated inhibition of the polyprotein processing activity of PLpro by UbV.CV2.1, whereas mutations in subsite-2 did not (**Fig 6C**). Overall, these results are consistent with a catalytic mechanism in which the polyprotein substrate binds to PLpro through interactions that include subsite-1 but not subsite-2, and consequently, interactions of the UbV.CV2.1 dimer with subsite-1 but not subsite-2 are required for inhibition of polyprotein processing by PLpro.

## Discussion

Even though vaccines against SARS-CoV-2 were quickly developed and have great efficacy against severe disease [32,33], a certain part of the population will be unable to receive such a prophylaxis due to, for example, being immunocompromised or being too young to be immunized. Furthermore, vaccines are rarely 100% effective [34,35] and the protective capacity of the induced immunity wanes over time [36,37], highlighting the continuing need for therapeutics for those who are suffering from severe symptoms of COVID-19, despite their vaccination status. SARS-CoV-2 PLpro is a promising drug target as it has proteolytic, deubiquitinating and deISGylating activities, the former being important for viral replication, and the latter two being important for viral innate immune evasion. Moreover, the variation in the PLpro-encoding sequence, amongst all known variants of concern of SARS-CoV-2, is minimal, which promises a broad activity for drugs targeting PLpro.

Indeed, substantial effort is being dedicated to the development of SARS-CoV-2 protease inhibitors [38,39], of which nilmatrelvir (used in Pfizer's drug combination Paxlovid) was

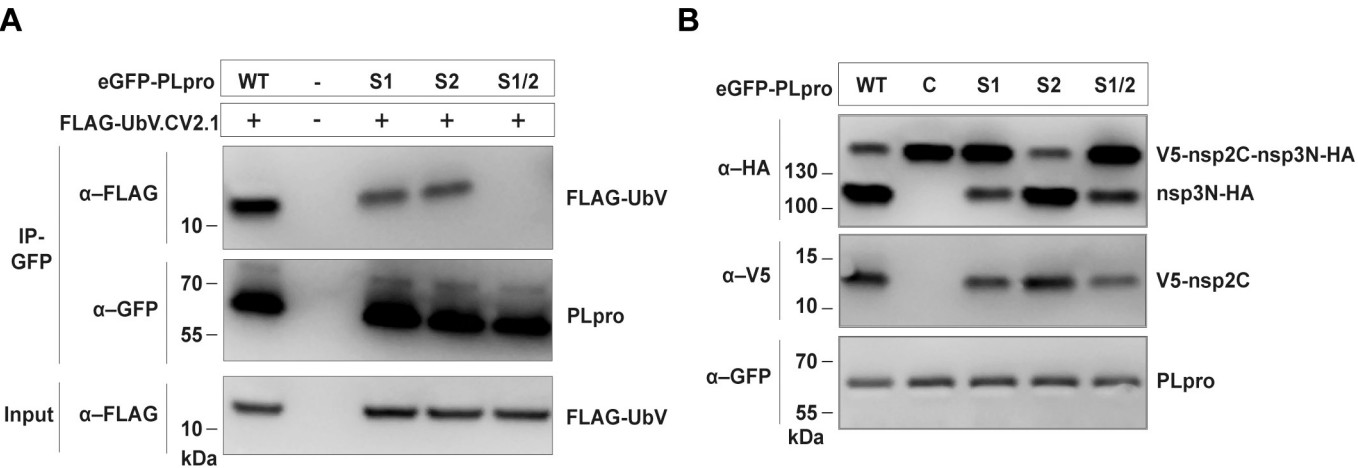

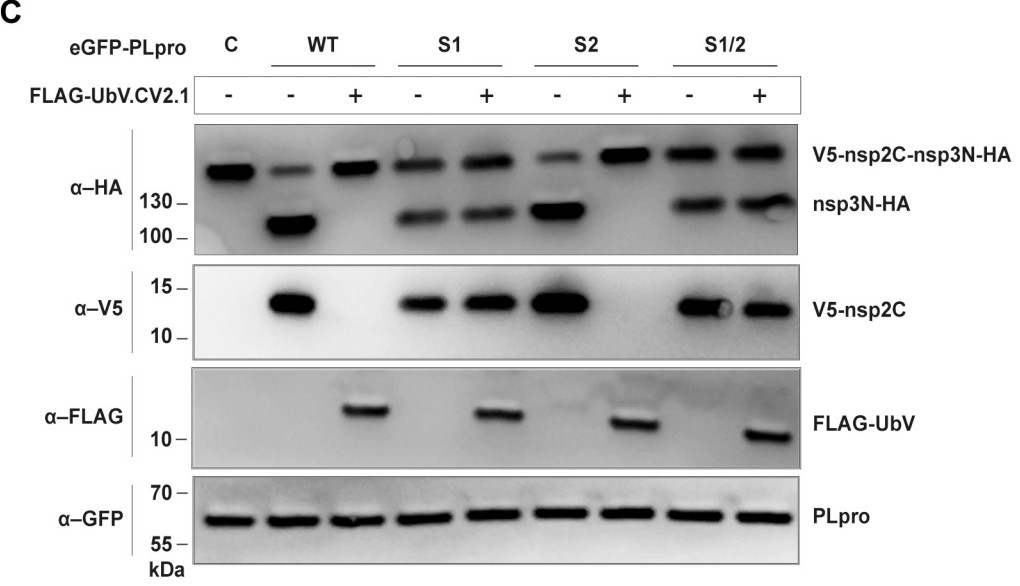

**Fig 6. Effects of PLpro mutations on binding and inhibition by UbV.CV2.1. (A)** FLAG-tagged UbV.CV2.1 was transcribed and translated *in vitro* with eGFP-tagged wt PLpro or PLpro variants containing alanine substitutions in subsite-1 (S1), subsite-2 (S2) or both subsites (S1/S2) (see **Fig 5C**). Proteins were immunoprecipitated with anti-GFP antibodies and immunoblotted using the indicated antibodies. **(B)** An *in vitro* protease assay was performed by transcribing and translating the indicated eGFP-tagged PLpro proteins together with an HA-nsp3N-nsp2C-V5 substrate. Western blotting with the indicated antibodies was used to examine proteolytic activity. The catalytically inactive C111A mutant 'C' was used as a negative control. **(C)** The protease assay described in (B) was performed in the presence or absence of FLAG-UbV.CV2.1 in the *in vitro* transcription/translation reaction, as indicated.

recently approved for use [40,41]. Most of the inhibitors developed thus far target the main protease, Mpro, and most of these are small chemical compounds that bind to a small region of the protease, likely providing the virus with possibilities for the evolution of drug resistance [42]. Various compounds have also been reported to inhibit PLpro, mainly based on high-throughput screens of clinically approved drugs or previously reported inhibitors of SARS-CoV PLpro, such as GRL-0617 [8,43–45]. However, a recent study invalidated some of these hits, and thus far, few have made it into clinical trials for treatment of COVID-19 [46,47]. The

need for continuous development of effective and innovative drugs that may keep current and future COVID patients safe from severe disease outcomes has therefore not been resolved.

Through screening of a large phage-displayed library, we have engineered UbVs that are able to bind the SARS-CoV-2 PLpro selectively and strongly through a relatively large molecular surface distal to the catalytic pocket. We believe UbV binding to this distal site sterically blocks the substrates and prevents their access to the catalytic site. This provides a promising approach for development of antiviral drugs that efficiently inhibit the multiple functions of the viral protease, thereby obstructing viral replication. Resistance may also be less likely to evolve due to the larger interaction surface, which leaves the virus with less possibilities for evolving effective combinations of mutations needed to repel the antiviral UbV without affecting the functions of the protease. This hypothesis will have to be investigated in future studies, as well as the feasibility of efficient delivery of these protein-based antiviral agents into infected cells *in vivo*. Many cell-penetrating peptides have been identified that could possibly support entrance of UbVs into the cell, such as TAT and arginine-rich peptides [48–50], with some showing favorable results *in vivo* [51]. A comparable approach was successfully applied for the delivery of Ub into cells [52], suggesting great promise that a similar technique could be applied for the therapeutic delivery of UbVs.

Our findings demonstrate that some UbVs actively inhibit the deISGylating and DUB activities of PLpro and thus affect the ways in which PLpro normally exhibits its innate immune evasive properties. Furthermore, the inhibitory effect of the UbV on the viral polyprotein cleavage in cells, during infection, drastically reduced the production of functional viral nsps. The release of viral nsps from the polyprotein precursors, among which is the viral RNA-dependent RNA polymerase more downstream, is necessary for efficient viral mRNA/genome production in the cell to produce more viral proteins and successfully kick-start the SARS-CoV-2 infection. However, in UbV-expressing cells, we believe that inhibition of PLpro's function in cleaving the N-terminal parts of the viral polyprotein likely also critically interferes with the rest of the processing of the polyproteins, which is normally executed by the main 3C-like protease in nsp5. This effect has been described in detail for distant relative equine arteritis virus [53], and although this has not been investigated thoroughly for the coronaviruses, it is likely that interfering with the concerted regulation of the polyprotein processing by PLpro and Mpro, in this case by inhibition of PLpro, will result in grossly interrupted production of functional nsp subunits, as we have observed in our study here. Upon expression of the UbV, production of viral RNA indeed was vastly diminished, and production of infectious progeny was consequently also greatly reduced by almost 5 orders of magnitude. This confirms that PLpro is a highly promising antiviral target, and UbVs can act as strong inhibitors of its critical functions.

An important consideration in the development of antiviral drugs is their safety in patients. As PLpro shares structural similarities with USPs [8,54,55], we assessed binding of the UbVs to various human USPs. Our results indicate that there is minimal binding of the UbVs to the human USPs. Furthermore, UbVs lack the two C-terminal glycines that are essential for Ub to be covalently attached to target proteins. Thus, UbVs will not interfere with the numerous cellular pathways that are regulated by ubiquitination. Our studies suggest that UbVs in general do not cause gross toxicity in cells, illustrated by our results showing that UbV expression does not affect general cellular tubulin levels and that Vero E6 cells stably expressing UbV.CV2.1a grew at a very similar rate compared to the parental cells and had identical morphology, exhibiting no evidence of cellular stress or toxicity.

In conclusion, we developed potent antiviral UbVs against SARS-CoV-2 PLpro. Their activity was assessed in biochemical and cellular assays, as well as their direct inhibitory effect on viral replication in cell culture. This, together with previous work on UbVs against

MERS-CoV PLpro [15], provides a proof of principle of the strong and selective antiviral effects of UbVs against coronaviruses.

## Materials and methods

### Cell and virus culture

Human embryonic kidney 293T cells (HEK293T, containing the SV40 T-antigen) were maintained in Dulbecco's modified Eagle's medium (DMEM, Lonza), supplemented with 10% fetal calf serum (FCS, Bodinco BV) with 100 units/mL penicillin, 100 units/mL streptomycin and 2 mM L-glutamine. VeroE6 cells were grown in DMEM with 10% FCS, 100 units/mL penicillin and 100 units/mL streptomycin. Polyclonal VeroE6 stable cells were selected in 10 μg/mL puromycin following infection with lentivirus and were cultured under selection using 1 μg/mL puromycin. All cell lines tested negative for mycoplasma.

Clinical isolate SARS-CoV-2/human/NLD/Leiden-0008/2020 (Leiden-0008) was isolated from a PCR-positive human throat swab and was passaged three times in Vero E6 cells. The spike protein of this isolate contains the D614G mutation. The complete genome sequence of this isolate is available under GenBank accession number MT705206.1 and the virus stock used contained only a low level of cell culture adaptive mutations compared to this consensus sequence, especially in the spike furin cleavage site region, where less than 2% sequence heterogeneity was established by deep cDNA sequencing of the used stock. In addition, no mutations in PLpro were found compared to the published sequence of the Wuhan strain. Isolate Leiden-0008 was propagated and titrated in Vero E6 cells and was used for infection experiments in a BSL-3 facility at the Leiden University Medical Center. An infectious SARS-CoV-2 (USA_WA1/2020 strain) was generated from the cDNA clone [56] and used to perform the infections for RT-qPCR and analysis of nsp1 expression at the Washington University School of Medicine.

### Antibodies

The following primary antibodies and dilutions were used for western blot analyses: mouse anti-FLAG (clone M2, #F1804, Sigma-Aldrich, diluted 1:2000), mouse anti-α-Tubulin (clone B-5-1-2, #T5168, Sigma-Aldrich, diluted 1:2000), mouse anti-β-Tubulin (clone AA2, #T8328, Sigma-Aldrich, diluted 1:10000), mouse anti-hISG15 (clone F-9, #sc-166755, Santa Cruz, diluted 1:400), mouse anti-V5 (clone 2F11F7, #37–7500, Thermo Fisher/Invitrogen, diluted 1:2000), rabbit anti-GFP (042150, Leiden, diluted 1:500), anti-FLAG M2-Peroxidase (HRP) (#A8592, Sigma-Aldrich, diluted 1:500), polyclonal anti-GFP (#A10260, Invitrogen, diluted 1:1000), mouse monoclonal anti-β-Actin (#12262S, Cell Signaling, diluted 1:10000), HRP-conjugated anti-V5 (#R961-25, Thermo Fisher, diluted 1:2500), mouse anti-HA (#H3663, Sigma Aldrich, diluted 1:1000), anti-SARS-CoV-2 nsp1 (diluted 1:300), anti-SARS-CoV nsp3 (DGD7, rabbit, diluted 1:500, [57]), rabbit anti-SARS-CoV M protein (EKU9, Leiden, rabbit, diluted 1:1000, [57]). Some of these primary antibodies were detected with biotin-conjugated goat anti-mouse IgG (#31802, Invitrogen, diluted 1:10.000), goat anti-mouse IgG HRP secondary antibody (#31430, Thermo Fisher, diluted 1:50000), goat anti-mouse IgG HRP (#P0447, Dako, diluted 1:2500) or biotin-conjugated donkey anti-rabbit IgG (#A16033, Thermo Fisher/Invitrogen, diluted 1:2000), and tertiary antibody Cy3-conjugated mouse anti-biotin (#200-162-211, Jackson, diluted 1:2500) was also used. Anti-nsp1 and anti-β-tubulin antibodies were detected using goat anti-human polyclonal (Invitrogen, diluted 1:1000) and goat anti-mouse HRP conjugate (BioRad, diluted 1:4000)

The following antibodies and corresponding dilutions were used for immunofluorescence analyses: mouse anti-FLAG (clone M2, #F1804, Sigma-Aldrich, diluted 1:1000), rabbit anti-

SARS-CoV M (EKU9, [58], diluted 1:1000), Alexa488-conjugated goat-anti-rabbit IgG (Invitrogen, diluted 1:300) and Cy3-conjugated donkey anti-mouse IgG (Jackson ImmunoResearch Laboratories, diluted 1:1000). Nuclei were stained with Hoechst 33258 (Thermo Fisher, diluted 1:100).

## Plasmid construction

Plasmids pET28a-LIC (#26094), VSV-G (#138479) and psPAX2 (#12260) were obtained from Addgene. We also obtain plasmids from other sources: pLVX-AcGFP-N1 (Takara #632154), pLVX puro (Takara #632164), pcDNA3.1/zeo(+) (Invitrogen V86020), pET53-DEST (NovoPro V010948). pLJM1 eGFP-1 plasmid was a kind gift from Jason Moffat at the University of Toronto.

Plasmids used for the polyprotein cleavage assay were generated as follows: DNA fragments encoding UbVs, SARS-CoV-2 PLpro, and SARS-CoV-2 polyprotein were obtained from Integrated DNA Technologies or Twist Bioscience and cloned into pET53, pHH0103 plasmids (for bacterial expression), pLJM1 eGFP-1, pLVX puro (for lentiviral transduction), or pcDNA 3.1/zeo(+) (for in-vitro transcription/translation) using Gibson assembly (NEB E2611) or via digestion/ligation methods. Constructs were verified by Sanger sequencing at the Centre for Applied Genomics (TCAG) facility at the University of Toronto.

For the deISGylation and DUB assays, a pcDNA3.1-SARS-CoV-2-PLpro-WT-V5 plasmid (amino acids 1564–1878) was constructed by PCR amplification of synthetic codon-optimized gBlock, cloned in-frame with a sequence encoding a V5-tag at the C-terminus into the pcDNA3.1(-) vector (Invitrogen). This plasmid was then used to produce the catalytically inactive mutant pcDNA3.1-SARS-CoV-2-PLpro-C-V5, using a Quickchange approach. Other plasmids used in this study were provided by others or described elsewhere: pLenti6.3-FLAG-Ub WT and pcDNA3.1-Ub.AA [15], pcDNA3.1-empty (Invitrogen), pCAGGS-V5-hISG15-GG and CS111-hHerc5-HA [59], and pCAGGS-HA-hUbE1L and pCMV2-FLAG-UbcH8 [60].

For expression and purification, the open reading frame encoding the SARS-CoV-2 PLpro protease domain was identified by aligning the translated sequence of the isolate Wuhan-Hu-1 (NC_045512) with the PLpro domain sequences from MERS-CoV and SARS-CoV [15,61]. The nucleotide sequence of the identified region was codon-optimized for expression in *E. coli* and synthesized (Integrated DNA Technologies). The synthetic DNA template was amplified by the PCR, and the amplicon was ligated into the bacterial expression vector pGEX-6P-1. Construct fidelity was confirmed by DNA sequencing at The Centre for Applied Genomics (Toronto, Ontario). As described, the resulting SARS-CoV-2 GST-PLpro fusion protein expression construct was used in phage-displayed UbV selections similar to those performed for the MERS-CoV PLpro enzyme [15].

To facilitate co-crystallization of SARS-CoV-2 PLpro bound to UbV.CV2.1, the codon encoding for the active site cysteine of the protease was mutated to a serine codon (C111S) [8,31,62] to generate a new expression plasmid based on pET24(b)+. Synthetic DNA (Integrated DNA Technologies) coding for the PLpro$^{C111S}$ protein was amplified by the PCR and ligated into the bacterial expression vector pET24(b)+. Construct fidelity was confirmed by DNA sequencing at The Centre for Applied Genomics (Toronto, Ontario). Replacement of the GST tag with an N-terminal His$_6$-tag improved yields of the PLpro$^{C111S}$ mutant for crystallization trials.

## Expression and purification of PLpro$^{C111S}$ and UbV.CV2.1

Active GST-PLpro fusion protein was expressed and purified by affinity chromatography using GST•Bind Resin (MilliporeSigma) followed by gel filtration chromatography based on previous procedures [15].

His$_6$-PLpro$^{C111S}$ was expressed from a pET24(b)+-based vector in *E. coli* BL21-Gold (DE3) cells (Stratagene). Transformed *E. coli* cells were grown aerobically in 100 ml of Luria–Bertani (LB) medium supplemented with 35 μg/ml kanamycin at 37˚C for ~16 h. Outgrowth cultures were prepared by inoculating 1 L of LB Medium supplemented with 35 μg/ml kanamycin to an optical density at 600 nm [OD$_{600}$] of ~0.1. Cultures were grown aerobically at 37˚C to late logarithmic phase (OD$_{600}$ between 0.8 and 0.9) (~2.25 h). His$_6$-PLpro$^{C111S}$ expression was then induced with 0.5 mM isopropyl-ß-D-1-thiogalactopyranoside (IPTG), and the cultures were grown at 16˚C for 16–18 h with shaking. Cells were harvested by centrifugation (3,440 x *g*) at 4˚C for 45 min and kept frozen at -80˚C until protein purification.

Cells were thawed, combined, and resuspended in ice-cold lysis buffer (50 mM TRIS [pH 7.5], 300 mM NaCl, 5 mM imidazole [pH 7.5], 50 μg/ml DNase) to a final volume of 80 ml. Cells were subsequently lysed using an Avestin EmulsiFlex C3 high-pressure cell homogenizer (ATA Scientific Instruments) and the lysate was clarified via high-speed centrifugation (17,200 x *g*) at 4˚C for 1 h. The supernatant containing soluble His$_6$-PLpro$^{C111S}$ protein was incubated end-over-end at 4˚C for 1 h with 5–6 ml of Ni–NTA Superflow resin (Qiagen), which was pre-equilibrated with the lysis buffer. The lysate-resin slurry was decanted into a 15-ml gravity flow column and washed with ~15 column volumes of lysis buffer, followed by ~4 column volumes of lysis buffer supplemented with 15 mM imidazole (pH 7.5), and ~4 column volumes of lysis buffer supplemented with 30 mM imidazole (pH 7.5). PLpro$^{C111S}$ protein was eluted in 15 ml lysis buffer supplemented with 250 mM imidazole (pH 7.5). The eluted protein was digested with HRV 3C PreScission Protease (GE Healthcare) to remove the His$_6$-tag and dialyzed against 2 L of dialysis/gel filtration buffer (20 mM TRIS [pH 7.5], 150 mM NaCl) at 4˚C for 16–18 h. His$_6$-free PLpro$^{C111S}$ was separated from HRV 3C PreScission Protease using a Superdex 75 gel filtration column (GE Healthcare). Fractions containing PLpro$^{C111S}$ were identified by SDS-PAGE analysis, and protein concentrations (mg/ml) were determined using a NanoDrop One (ThermoFisher) by measuring the absorbance at λ = 280 nm and using the theoretical mass (36.03 kDa) and extinction coefficient ($\varepsilon_{280}$ = 45,270 M$^{-1}$ cm$^{-1}$) of PLpro$^{C111S}$. Fractions containing purified PLpro$^{C111S}$ were pooled and concentrated down to 1–2 ml at 4˚C using a 10 kDa molecular weight cut-off (MWCO) centrifugal filter unit.

UbV.CV2.1 protein was expressed and purified identically to the His$_6$-PLpro$^{C111S}$ protein, except that 150 μg/ml ampicillin was used as the selection pressure for *E. coli* transformants and the eluted protein was not digested with HRV 3C PreScission Protease. Protein concentrations were determined using a NanoDrop One by measuring the absorbance at λ = 280 nm and using the theoretical mass (11.81 kDa) and extinction coefficient ($\varepsilon_{280}$ = 4470 M$^{-1}$ cm$^{-1}$) of UbV.CV2.1.

## Co-crystallization of PLpro$^{C111S}$ and UbV.CV2.1

A 1:6 molar ratio of purified PLpro$^{C111S}$ to UbV.CV2.1 was prepared in gel filtration buffer supplemented with 5 mM DTT and incubated at 4˚C for 16–18 h. The resulting PLpro$^{C111S}$–UbV.CV2.1 complex was purified using a Superdex 75 gel filtration column. Fractions containing the complex were identified by SDS-PAGE analysis, and concentrations were determined using a NanoDrop One by measuring the absorbance at λ = 280 nm and using the theoretical mass (47.84 kDa) and extinction coefficient ($\varepsilon_{280}$ = 49,740 M$^{-1}$ cm$^{-1}$) of the complex. Fractions containing the purified complex were pooled, supplemented with 5 mM TCEP (pH 8.0), and concentrated to ~15 mg/ml at 4˚C using a 10-kDa MWCO centrifugal filter unit.

The PLpro$^{C111S}$–UbV.CV2.1 complex was crystallized at 14.9 mg/ml using the hanging drop vapor diffusion method in a condition composed of 20% PEG 3350, 0.2 M sodium bromide, 0.1 M BIS-TRIS propane (pH 6.3), 50 mM lithium chloride. Crystals appeared after 5

days of incubation at 8˚C. Prior to X-ray data collection, a single crystal was swept through cryoprotectant composed of the initial crystallization condition supplemented with 30% glycerol and subsequently flash-frozen in liquid nitrogen.

### X-ray data collection and structure determination

X-ray diffraction data were collected in-house at 100 K using a Rigaku MicroMax HF X-ray generator and R-AXIS IV++ image plate detector. Data were indexed and integrated using MOSFLM [63] and scaled using Aimless [64] within the CCP4i2 program suite [65]. Structure solution was performed by molecular replacement using PHASER [66] within the PHENIX crystallography suite [67]. For the molecular replacement, coordinates for chain B of PDB 7CJM [62] were used as a model of the SARS-CoV-2 PLpro, and a SWISS-MODEL model [68] was generated for UbV.CV2.1, using its sequence. Molecular replacement proceeded smoothly except that we found 4 molecules of PLpro but 8 molecules of the UbV. In addition, the resulting electron density indicated that in each UbV, a continuous strand of electron density followed strand 1 into the neighbouring UbV, rather than the formation of a β-turn (**S4 Fig**). This indicated the presence of a strand-swapped UbV dimer, even though the molecular replacement model for the UbV was monomeric. Following molecular replacement, iterative cycles of phenix.refine [67] and manual remodelling of the molecular model using the molecular graphics software Coot [69] were performed until Rwork and Rfree converged; we obtained final values of 22.5 and 27.0% (see Table 1) with no Ramachandran violations. Group B-factors were used and TLS refinement [70] was not used due to the low resolution of the data.

### Octet Bio-Layer Interferometry (BLI)

Experiments to determine the affinity and binding kinetics of UbVs for the SARS-CoV-2 PLpro were performed as described [15,71]. BLI experiments were performed on an Octet HTX instrument (ForteBio) using anti-GST antibody biosensors for GST-tagged ligands and His-tagged analytes at 1000 rpm and 25˚C. Concentrated analyte and ligand proteins were diluted into BLI reaction buffer (25 mM HEPES pH 7.0, 150 mM NaCl, 0.1 mg/ml bovine serum albumin and 0.01% Tween 20). GST-fused PLpro was first captured on GST biosensors (ForteBio) from a 2 µg/mL solution in PBT, followed by a 180 s quench step with 100 µg/mL biotin. After equilibrating with PBT, loaded biosensors were dipped for 600 s into wells containing serial 3-fold dilutions of UbV and subsequently were transferred back into assay buffer for 600 s dissociation. Sensorgram binding response raw data were reference subtracted and were fitted with 1:1 binding model using ForteBio's Data Analysis software 9.0. Three replicates were measured for each UbV and the average and coefficient of variance were determined. Binding constants $K_D$ were obtained by fitting the response wavelength shifts in the steady-state regions using single-site binding system.

### Binding specificities of the UbVs

The binding specificities of the UbVs towards a panel of DUBs were assayed by protein ELISA. DUB proteins were immobilized on a plate at 2 ug/mL and 300 nM Flag-UbV was added. Bound UbV protein was detected by the addition of anti-FLAG HRP antibody and color was developed after the addition of TMB peroxidase substrate. The absorbance of the plate was read at 450 nm, and the mean of the absorbance was represented in a heat map on a white (minimum value)–black (maximum value) gradient. For Ub.ΔGG, mean of the absorbance was normalized to the signal in UbV.CV2.1 to reflect the relative binding specificity between wt Ub and the SARS-CoV-2 PLpro-specific UbV.

## ISG15-7-amido-4-methylcoumarin (AMC) deconjugation assay

Inhibition assays using ISG15-AMC (both Boston Biochem) as substrate were performed as described [15,28,72]. Assays were performed at 37°C in assay buffer (50 mM HEPES, pH 7.5, 0.01% Tween 20, 1 mM dithiothreitol (DTT)) containing 1 mM ISG15-AMC substrate, three-fold serial dilutions of UbV, and 2.5 nM PLpro, which is the lowest concentration that showed linear activity in the first 10–15 min. PLpro and UbV were mixed in assay buffer and incubated for 10 min before the addition of the substrate. Activity was measured by monitoring the increase of AMC fluorescence emission at 460 nm (excitation at 360 nm) for 60 min using a BioTek Synergy2 plate reader (BioTek Instruments, Winooski, VT). $IC_{50}$ values were calculated using the GraphPad Prism8 software with the built-in equation formula (non-linear regression curve).

## Cellular deISGylation and DUB assays

For the deISGylation assay, HEK293T cells were transfected with pCAGGS-V5-ISG15-GG, pCAGGS-HA-hUbE1L, pCMV2-FLAG-UbcH8 and CS111-hHerc5-HA to visualize ISGylation of cellular substrates. These plasmids were then co-transfected with different combinations of the following plasmids: pcDNA3.1-SARS-CoV-2-PLpro-WT-V5, pcDNA3.1-SARS-CoV-2-PLpro-C-V5, and pcDNA3.1-UbVs. pcDNA3.1-empty vector was used to supplement to an equal amount of DNA transfected per well. Transfection were performed using the polyethylenimine (PEI 25K, Polysciences) transfection method. Briefly, per 2 μg of plasmid DNA, 6 μL of PEI was diluted in Opti-MEM medium (Lonza). The plasmid DNA was also diluted in Opti-MEM. Diluted PEI was then added to the diluted DNA, vortexed and incubated for 20 min. The mixture was then added to the cells (12-well plate, $3x10^5$ cells/well) in a dropwise manner and the cells were incubated at 37°C, 5% CO2. Cells transfected with the deISGylation assay were lysed 48 hours post-transfection in 2x Laemmli Sample Buffer (LSB, 250 mM Tris-base (pH 6.8), 4% SDS, 20% glycerol, 10 mM DTT, 0.01% Bromophenol blue). This assay was performed in a 12-well format.

For the DUB assay, HEK293T cells were transfected with different combinations of the following plasmids: pLenti6.3-FLAG-Ub WT, pcDNA3.1-SARS-CoV-2-PLpro-WT-V5, pcDNA3.1-SARS-CoV-2-PLpro-C-V5, and pcDNA3.1-FLAG-UbVs. pcDNA3.1-empty vector was used to supplement to an equal amount of DNA transfected per well. Transfections were performed as described above. Twenty-four hours post-transfection, the medium was aspirated and the cells lysed using 2x Laemmli Sample Buffer.

## SARS-CoV-2 PLpro *in vitro* protease assay

Protease assay was performed as described [73], with some modifications. DNA fragments encoding for SARS-CoV2 V5-nsp2C-nsp3N-HA were obtained as gBlocks from Integrated DNA Technologies and cloned into pcDNA3.1 zeo(+) plasmid, downstream of CMV and T7 promoters. *In vitro* transcription and translation were carried out using TNT quick coupled transcription/translation system-T7 (Promega L1170) following the manufacturer's instructions. To test for inhibition of SARS-CoV2 PLpro protease activity, *in vitro* transcription/translation was performed in the presence of purified $His_6$-FLAG-UbV.CV2.1a or $His_6$-FLAG-UbV.CV2.1b at a final concentration of 0.1 μM, 1.0 μM, or 10.0 μM while the negative control $His_6$-FLAG-Ub.ΔGG was used at the highest concentration of 10.0 μM. Reaction products were analyzed by western blotting using the following antibodies: HRP-conjugated V5 monoclonal antibody, mouse monoclonal anti-HA antibody, goat anti-mouse IgG HRP secondary antibody and mouse monoclonal anti-FLAG M2-peroxidase HRP antibody.

## SARS-CoV-2 PLpro protease assay in cell culture

Protease assay in cell culture was performed as described [15]. To evaluate the *in trans* cleavage activity of SARS-CoV-2 PLpro in the presence of UbV, HEK293T cells were co-transfected with plasmids encoding V5-nsp3C-nsp4N-HA (0.25 µg), eGFP-tagged SARS-CoV2 PLpro (0.25 µg), FLAG-tagged UbVs (0.5; 0.75 or 1 µg, as indicated) using Lipofectamine 2000 (Invitrogen 11668019). Empty pcDNA vector was added to supplement to a total of 1.5 µg of plasmid DNA transfected per well of a 6-wells cluster. 48-hour post-transfection, cells were lysed in lysis buffer (50mM Tris-HCl pH 7.4, 250mM NaCl, 2mM EDTA, 1.0% Triton X-100, 10% glycerol, 1x proteinase inhibitor) (Sigma-Aldrich, #11873580001)) for 30 min with end-to-end rotation at 4˚C.

## Western blot analysis

For the western blot analysis of the SARS-CoV-2 infection experiment, proteins were separated on a 15% SDS-PAGE gel, after which gels were blotted onto a 0.2 µm Amersham Hybond P PVDF western blotting membrane (Cytivia/Merck) and blocked in 1x casein (#C5890, Sigma-Aldrich) for 1h. Membranes were stained overnight with anti-SARS-CoV M antibody, as well as anti-GFP and anti-FLAG antibodies to visualize the GFP- and UbV.CV2.1a expression of the VeroE6 cells, respectively, and anti-alpha tubulin antibody as a loading control. The membranes were washed 3 times with PBS containing 0.05% Tween-20 (PBST) and incubated for 1 h with the secondary antibody in 0.5x casein. After washing again with PBST, the membranes were incubated in the dark with the tertiary antibody in 0.5x casein for 1 h. The membranes were washed in PBST and PBS, and visualized. Similar technique was used for the detection of SARS-CoV-2 nsp3, using an anti-nsp3 antibody, as well as anti-alpha-tubulin as controls.

For both the deubiquitination and the deISGylation assays, proteins were separated on an 8% and a 15% SDS-PAGE gel in parallel. The former was used to visualize the ubiquitin- or ISG15-conjugated protein smears, and the latter to visualize the loading control (α-Tubulin), SARS-CoV-2 PLpro-V5 and FLAG-UbVs. Gels were handled as described above until the secondary antibody step. Instead, the blots were incubated with goat anti-mouse HRP in 0.5x casein and visualized using the Bio-Rad Clarity Western ECL Substrate (Bio-Rad, #170–5061).

For the polyprotein cleavage assay, proteins were visualized using Pierce ECL western blotting substrate (Thermo Fisher, #32106) and Microchemi 4.2.

To check the expression of nsp1 in infected cells, VeroE6 cells stably expressing UbV. CV2.1a were seeded into 24-well plates at $1 \times 10^5$ cells/well and incubated overnight. The next day, cells were infected with SARS-CoV-2 and incubated at 37˚C. At 48 hpi, the supernatant was removed, and cells were treated with 1% SDS for 20 min to inactivate the virus. The samples were collected, equal amounts of protein were resolved by SDS-PAGE, and proteins were transferred to PVDF membranes. Western blots were probed with antibodies against nsp1 and the housekeeping protein β-tubulin in blocking buffer by incubating overnight at 4˚C. After washing in TBST, membranes were incubated for 1 h at room temperature with the respective secondary antibodies. The blot was developed with HRP substrate (Immobilon, Millipore) and imaged using iBright 1500, Invitrogen.

## Immunofluorescence assay (IFA)

VeroE6 cells expressing UbV.CV2.1a or parental VeroE6 cells were seeded onto coverslips from sustained culture, infected with SARS-CoV-2, and fixed for at least 4 h in 3% paraformaldehyde at 48 hpi. IFAs were performed as described previously [57]. Briefly, the coverslips were then washed with PBS containing 10 mM glycine. To permeabilize the cells, the

coverslips were incubated with PBS containing 0.1% Triton X-100 for 10 min. The coverslips were then washed in PBS before incubation with primary antibody in PBS with 2% FCS for 1 h. After incubation, the coverslips were washed again with PBS before incubation with secondary antibody in PBS/2% FCS for 1 h. The coverslips were mounted onto microscope slides with ProLong Glass antifade mounting fluid (Thermo Fisher/Invitrogen). Results were assessed using the Leica DM6B fluorescence microscope and LASX software (Leica Microsystems B.V.).

### SARS-CoV-2 infection experiments

For RT-qPCR, VeroE6 cells stably expressing UbV were seeded in 24-well plates ($1x10^5$ cells/ well) and incubated overnight. These and wt VeroE6 cells, which are both naturally susceptible to infection, [57] were infected with the SARS-CoV-2 USA_WA1/2020 strain generated by reverse genetics at a multiplicity of infection (MOI) of 0.01 in complete DMEM medium supplemented with 2% FBS and incubated at 37˚C [56]. At 24 hpi, the supernatant was collected to infect wt VeroE6 cells, and the cells were used to extract the total RNA. The infected stable and wt VeroE6 cells were collected in Trizol reagent and proceeded with RNA extraction using RNeasy (Qiagen) kit. cDNA was synthesized using Superscript III Reverse transcriptase (Invitrogen) and stored at -20˚C. The quantitative PCR (qPCR) was performed on StepOnePlus Real-Time PCR system (ABI) using SYBR Green master mix. Standard curve was plotted by serial 10-fold dilutions of synthesized cDNA from N-gene RNA of SARS-CoV-2, as described [74]. Briefly, the concatenated segments of the N gene were synthesized in a gBlocks fragment (Integrated DNA Technologies) and the RNA was generated using linearized PCR-II topo vector by *in vitro* T7-DNA-dependent RNA transcription. Viral RNA levels were measured by qPCR and normalized to an endogenous β-actin control. The following primers were used for N-gene (L Primer: ATGCTGCAATCGTGCTACAA, R primer: GACTGCCGCCTCTGCTC) and β-Actin amplification (L-primer: CTGGCACCCAGCACAATGA, R-primer: AAGTCATAGTCCGCCTAGAAGC). Data was processed using Prism software (GraphPad Prism 9.0). All experiments were done three times independently. Work using live virus was performed inside biosafety cabinets at the Biosafety Level 3 laboratories located at Washington University, School of Medicine, USA.

To determine the reduction in viral progeny upon UbV expression, VeroE6 cells, as well as the stable cell lines VeroE6-GFP and VeroE6-UbV.CV2.1a, were infected with SARS-CoV-2 at a MOI of 0.01 or 1. A clinical isolate of SARS-CoV-2 was used (SARS-CoV-2/human/NLD/ Leiden-0008/2020), isolated at the Leiden University Medical Center. Virus was diluted in Eagle's Minimal Essential Medium (EMEM, Lonza) containing 2% FCS, 100 units/ml penicillin and 100 units/ml streptomycin to reach the appropriate MOI. The different VeroE6 cells were seeded into 12-well plates the day before infection, including wells containing coverslips. On the day of infection, medium from the wells was replaced with SARS-CoV-2 virus dilutions and incubated on a shaker for 1 h at 37˚C. Inoculum was then removed, the cells were washed with PBS, and medium (EMEM, 2% FCS and antibiotics) was added to the wells. The cells were incubated at 37˚C. At 24 or 48 hpi, supernatants were collected for subsequent viral titration, cells were lysed in 4x LSB, and coverslips were fixed in 3% PFA. Lysates and coverslips were processed as described above.

To determine SARS-CoV-2 titres, the supernatants were subjected to plaque assay on VeroE6 cells. Experiments were performed independently 3 times and plaque assays were performed in duplicate for each sample with similar results. Briefly, $10^{-1}$ to $10^{-6}$ dilutions of supernatants were made in EMEM/2% FCS and added to the appropriate wells. This was then incubated for 1 h at 37˚C after which inoculum was removed and Avicel-containing semi-

solid overlay was added to the wells. Plates were incubated for 3 days at 37˚C, before fixing the cells and staining with crystal violet. Corresponding graphs were created using Graphpad Prism. Work using live virus was performed inside biosafety cabinets at the Biosafety Level 3 laboratories located at the Leiden University Medical Center, The Netherlands.

## Supporting information

**S1 Fig. Proteolytic removal of GST from purified SARS-CoV-2 PLpro.** Purified GST-SARS-CoV-2 PLpro fusion protein has a mass of ~60 kDa according to SDS-PAGE and could be readily cleaved into free GST (~25 kDa) and SARS-CoV2 PLpro (~35 kDa) using HRV3c Precision Protease. The lane on the far left contains the molecular weight ladder and masses are shown on the left. Remaining lanes are various optimizations of the cleavage assay, with optimal cleave results shown in the far right lane.
(TIF)

**S2 Fig. Octet Bio-Layer Interferometry (BLI) curves.** BLI curve fits of soluble UbVs with immobilized SARS-CoV-2 PLpro. Curves are shown for the parent UbV.CV2.1 and some of its optimized variants.
(TIF)

**S3 Fig. UbVs inhibit the deISGylation activity of SARS-CoV-2 PLpro.** Inhibition of SARS-CoV-2 PLpro by the cognate UbVs shown as dose-response curves using ISG15-AMC as a substrate. The $IC_{50}$ value was determined as the concentration of UbV that reduced enzyme's activity by 50%.
(TIF)

**S4 Fig. UbVs inhibit the DUB activity of SARS-CoV-2 PLpro.** Inhibition of the deubiquitinating activity of SARS-CoV-2 PLpro by UbVs, visualized by co-transfection of plasmids encoding FLAG-Ub.WT, SARS-CoV-2 PLpro-V5 (wildtype or catalytic mutant 'C') and FLAG-UbVs into HEK293T cells. Lysates were collected 24 hours post transfection and were subjected to western blot analysis. DUB activity of PLpro is shown by the removal of FLAG-Ub from cellular substrates. Co-expression of some of the UbVs causes inhibition of PLpro DUB activity and ubiquitination of cellular proteins.
(TIF)

**S5 Fig. Schematic representation of utilized constructs. (A)** Illustration of the construct used for the in vitro cleavage assay as shown in Fig 3A. The construct spans part of nsp2, part of nsp3 including the PLpro domain, as well as tags on either side for easy identification. **(B)** Representation of the constructs used for the polyprotein cleavage assay shown in Fig 3B. The construct spans part of nsp3 (excluding the PLpro domain), and part of nsp4, as well as tags on either side.
(TIF)

**S6 Fig. UbVs inhibit proteolytic activity of SARS-CoV-2 PLpro from the alpha and beta variants.** *In vitro* proteolytic cleavage capability of SARS-CoV-2 PLpro from the alpha (**A**) or beta (**B**) variants was assessed in the presence of the UbV.CV2.1a and 1b at different concentrations. Construct encoding N-terminal V5-tagged and C-terminal HA-tagged nsp2C-nsp3N (including the PLpro domain), V5-nsp2C-nsp3N-HA, was transcribed and translated *in vitro* in the presence of UbV.CV2.1a or 1b (with increasing doses) for 2 hours. Proteolytic cleavage activity was assessed by western blotting to detect the presence of N-terminal V5-tagged nsp2C and C-terminal HA-tagged nsp3N cleavage products.
(TIF)

**S7 Fig. Omit map demonstrating that UbV.CV2.1 forms a swapped dimer in the presence of the SARS-CoV2 PLpro B.** Electron density from a simulated annealing omit map generated by omitting residues 7–11 of all the UbV chains (chains E-L) from the model. The simulated annealing omit map was generated using Phenix (ref), electron density then was plotted up to 2.5 A around the UbVs (chain E and I of the model), and contoured at sigma = 1.2. The resulting electron density in chains E and I around this region shows 2 continuous beta strands through this region, indicative of a swapped dimer.
(TIF)

## Acknowledgments

We thank C. Tait-Burkard, R.C.M Knaap and E.J. Snijder for useful discussions.

## Author Contributions

**Conceptualization:** Vera J. E. van Vliet, Jacky Chung, Brian L. Mark, Marjolein Kikkert, Sachdev S. Sidhu.

**Formal analysis:** Vera J. E. van Vliet, Nhan Huynh, Alex Singer.

**Funding acquisition:** Jacky Chung, Brian L. Mark, Marjolein Kikkert, Sachdev S. Sidhu.

**Investigation:** Vera J. E. van Vliet, Nhan Huynh, Judith Palà, Ankoor Patel, Alex Singer, Cole Slater, Mariska van Huizen, Joan Teyra, Shane Miersch, Gia-Khanh Luu, Wei Ye, Nitin Sharma, Safder S. Ganaie, Raquel Russell, Chao Chen, Mindy Maynard.

**Methodology:** Vera J. E. van Vliet, Nhan Huynh, Jacky Chung, Mariska van Huizen, Nitin Sharma.

**Project administration:** Jacky Chung, Brian L. Mark, Marjolein Kikkert, Sachdev S. Sidhu.

**Software:** Alex Singer.

**Supervision:** Jacky Chung, Gaya K. Amarasinghe, Brian L. Mark, Marjolein Kikkert, Sachdev S. Sidhu.

**Visualization:** Vera J. E. van Vliet, Nhan Huynh, Alex Singer.

**Writing – original draft:** Vera J. E. van Vliet, Nhan Huynh, Ankoor Patel, Alex Singer, Cole Slater, Jacky Chung, Gaya K. Amarasinghe, Brian L. Mark, Marjolein Kikkert, Sachdev S. Sidhu.

**Writing – review & editing:** Vera J. E. van Vliet, Alex Singer, Jacky Chung, Nitin Sharma, Gaya K. Amarasinghe, Brian L. Mark, Marjolein Kikkert, Sachdev S. Sidhu.

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
