## [Decision Letter · Decision Letter 0]

26 Oct 2022

Dear Ms van Vliet,

Thank you very much for submitting your manuscript "Ubiquitin variants potently inhibit SARS-CoV-2 PLpro and viral replication via a novel site distal to the protease active site" for consideration at PLOS Pathogens. As with all papers reviewed by the journal, your manuscript was reviewed by members of the editorial board and by several independent reviewers. The reviewers appreciated the attention to an important topic. Based on the reviews, we are likely to accept this manuscript for publication, providing that you modify the manuscript according to the review recommendations.

Your manuscript was reviewed by three experts in the field. All of the reviewers were enthusiastic about the manuscript and felt that it was an important contribution. Please respond to the comments of the reviewers, providing additional data as is feasible in a short time frame.

Sincerely,

Stanley Perlman

Associate Editor

PLOS Pathogens

Ron Fouchier

Section Editor

PLOS Pathogens

Kasturi Haldar

Editor-in-Chief

PLOS Pathogens

orcid.org/0000-0001-5065-158X

Michael Malim

Editor-in-Chief

PLOS Pathogens

orcid.org/0000-0002-7699-2064

Your manuscript was reviewed by three experts in the field. All of the reviewers were enthusiastic about the manuscript and felt that it was an important contribution. Please respond to the comments of the reviewers, providing additional data as is feasible in a short time frame.

Reviewer Comments (if any, and for reference):

Reviewer's Responses to Questions

**Part I - Summary**

Reviewer #1: This manuscript from van Vliet and colleagues describes the identification of a variant of ubiquitin (UbV.CV2.1a) that binds to SARS-CoV-2 PLpro and inhibits enzymatic activity. They selected this variant from a phage display library and found that the purified UbV binds to purified PLpro and inhibits enzymatic activity. They obtained a crystal structure of SARS-CoV-2 PLpro in complex with a UbV variant and found that the UbV bound distal to the active site of the enzyme, in an unexpected dimer configuration (Figure 4). Using Vero E6 cells stably expressing UbV.CV2, they found that replication and production of infectious virus was reduced by several logs as compared to the control cell lines at 24 and 48 hours post infection. Overall, this is a very interesting and potentially important study as it identifies sites distal to the PLpro catalytic site that may be targeted for antiviral drug development. The paper is well written and documents well controlled experiments.

Reviewer #2: In this manuscript the authors develop ubiquitin variants against SARS-CoV-2 PLpro as a means to inhibit infection by blocking proteolytic activity of PLpro. The Nsp3 protein of SARS-CoV-2 in general and PLpro in particular are critical components conferring the ability of viral spread and immune evasion. This study is therefore an important contribution to the field. General and specific comments on the article are provided below:

General comments:

The study is well-conducted with in vitro experiments combined with cell-based experiments and also in those infected with live virus. The reduction in viral titres in the presence of the UbV is significant. However, while the authors describe these Ub variants as potential antiviral therapeutic molecules, it is not clear how efficiently they can be delivered to cells. I have very few criticisms on this study; only a few minor comments.

Specific comments:

Given that the binding of the UbV appears to be equivalent for PLpro from CoV1 and CoV2, it might be useful to show whether CoV1 is inhibited as well.

Along the same lines, the activity of CoV2 and CoV1 PLpro have been shown to be different in their ability to hydrolyse ubiquitin modifications and ISG15 modifications (Munnur et al Nat Imm 2021, Shin et al Nature 2020). The loss of ISGylation (but much less so for ubiquitylation) is very easily visible in lysates from SARS-CoV2-infected or PLpro expressing cells, which has implications in immune subversion, even in the absence of a productive viral lifecycle (Munnur et al, Nat Imm 2021). It would therefore be important and easy to measure bulk ubiquitylation and ISGylation in these cells expressing the Ub variants and PLpro (and IFN-treated) or infected with virus, to test whether these profiles are affected.

Similarly, only inhibition of the deISGylase activity of PLpro has been shown with the UbVs; could the authors show whether the UbVs also affect DUB activity of the PLpro?

Proteolytic activity of PLpro is affected to same extent in PLpro mutant and UbV.CV2.1a/b (Fig 2A and 2B) - how does the distal binding of UbVs to PLpro impart the same level of inhibition as point mutation in active site for PLpro mutant – can authors speculate using structural information?

It would be useful if the authors could comment on whether these variants would be able to equally inhibit PLpro from later viral variants of concern, based on the evolution of mutations in PLpro and their predicted structures.

Minor comments:

- Figure S1 needs a ladder on the gel

- Fig 2C – can the expression level of Ub.AA and UbV be shown

- Please include β-strands in Figure for easy identification of strands

- The authors state that infection was performed in Vero cells. It should be clarified whether these are Vero cells expressing Tmprss2 or Ace2 to make them susceptible to infection.

Reviewer #3: Van Vliet and colleagues have made use of phage display technology to identify a ubiquitin variant that binds to the SARS-CoV-2 protease PLPro. This manuscript characterized the binding, mode of action of the inhibitor and even crystalized the protein in complex with PLpro. Furthermore, they showed that transfection of the construct into cells was sufficient to reduce viral titers by five logs. They show that this is due to inhibition of PLpro’s cleavage activity of viral proteins thus preventing their activity and the formation of new viral particles but also based on inhibition of ISG15 deconjugation which is a mechanism of immune escape. The manuscript is thorough, well controlled and impactful given the need for complementary chemical inhibition strategies for the immunocompromised. I have one suggestion which I think will bolster and broaden their findings.

**Part II – Major Issues: Key Experiments Required for Acceptance**

Reviewer #1: 1. Figure 3 documents the reduction in viral RNA and pfu/ml at 24 and 48 hours post-infection and western blots showing the absence of M protein in UbV expressing cells. The results are consistent with UbV.CV2.1a impairing virus replication, but this does not show the target of this inhibition in the virus infected cells. One way to do this would be to infect cells at a very high MOI (perhaps using Vero-ACE2-TMPRSS2 expressing cells) and look at early times after infection (3-4 hrs) for the production of the ORF1a polyprotein products by western blot. The 3CLpro should be active and cleaving the downstream region of the polyprotein, but in the presence of the Ub variant, PLpro would be inhibited, generating a nsp1-2-3 polyprotein. This experiment would directly test if UbV.CV2.1a is directly blocking processing by PLpro . Alternatively, the Ub variant might be affecting virus entry or translation in some unexpected way, so it is important to determine if SARS-CoV-2 can enter and translate RNA into the ORF1a polyprotein in the presence of UbV.CV2.1a expression. Alternatively, this issue about what exactly UbV.CV2.1a is targeting in virus-infected cells could be part of the discussion.

Reviewer #2: see comments

Reviewer #3: The authors hypothesize that their inhibitor will affect endogenous ISGylation based on PLPro’s activity as a protease, however they test this in Vero cells which are derived from monkey kidney and have an aberrant interferon response. In addition, unlike ubiquitin ISG15 is not well conserved between metazoan species. I suggest that the authors broaden their work to test the activity of their inhibitor in a human cell line which will express ISG15 and be conjugation competent (since not all cell lines express Ube1L the E1 enzyme for ISG15). The authors could use A549 which ectopically express Ace2 or Calu-3 or even THP-1 cells. With these cell types they could look for the loss of targeting of ISGylation following ectopic expression of the ubiquitin variant or reduced viral load in the lung epithelial cell lines. This could also be done using ISGylated targets taken from cells and subjected to PLPro activity in vitro, which may have more complex chain topology than ISG15-AMC. Ultimately, these experiments will allow them to test inhibition of human ISGylation in lung epithelial cells, which will be more relevant for the implications of their findings.

**Part III – Minor Issues: Editorial and Data Presentation Modifications**

Reviewer #1: 1. The discussion focuses on the potential of Ub variants as inhibitors. However, a second important observation from this study is that blocking a site distal to the active site inhibits PLpro activity, thus identifying an additional target for antiviral drug discovery.

2. Short title: modify to something like Ubiquitin variants identify distal sites for inhibiting SARS-CoV-2 PLpro activity. More experiments using soluble UbVs that can be administered to cells would be needed to say UbVs are antiviral agents. It does seem like this is a future goal for this research.

3.line 649: which isolate of SARS-CoV-2? Was it passaged and re-sequenced (it was it adapted to replication in VeroE6 cells). This may be important later if you start to look at inhibition of VOCs using this system.

Reviewer #2: see comments

Reviewer #3: (No Response)

PLOS authors have the option to publish the peer review history of their article (what does this mean?). If published, this will include your full peer review and any attached files.

Reviewer #1: No

Reviewer #2: **Yes: **Sumana Sanyal

Reviewer #3: No

Figure Files:

Data Requirements:

Reproducibility:

References:

---

## [Editor Report · Decision Letter 1]

13 Dec 2022

Dear Ms van Vliet,

We are pleased to inform you that your manuscript 'Ubiquitin variants potently inhibit SARS-CoV-2 PLpro and viral replication via a novel site distal to the protease active site' has been provisionally accepted for publication in PLOS Pathogens.

Best regards,

Stanley Perlman

Academic Editor

PLOS Pathogens

Ron Fouchier

Section Editor

PLOS Pathogens

Kasturi Haldar

Editor-in-Chief

PLOS Pathogens

orcid.org/0000-0001-5065-158X

Michael Malim

Editor-in-Chief

PLOS Pathogens

orcid.org/0000-0002-7699-2064
---

## [Editor Report · Acceptance letter]

17 Dec 2022

Dear Ms van Vliet,

We are delighted to inform you that your manuscript, "Ubiquitin variants potently inhibit SARS-CoV-2 PLpro and viral replication via a novel site distal to the protease active site," has been formally accepted for publication in PLOS Pathogens.

Best regards,

Kasturi Haldar

Editor-in-Chief

PLOS Pathogens

orcid.org/0000-0001-5065-158X

Michael Malim

Editor-in-Chief

PLOS Pathogens

orcid.org/0000-0002-7699-2064